# Study on the Spatial and Temporal Evolution Patterns of Green Innovation Efficiency and Driving Factors in Three Major Urban Agglomerations in China—Based on the Perspective of Economic Geography

**Biao Hu [1], Kai Yuan [1,\*], Tingyun Niu [2], Liang Zhang [3] and Yuqiong Guan [4]**

[1] School of Management, Tianjin University of Technology, Tianjin 300384, China; hubiaomail@126.com
[2] School of Public Affairs, Xiamen University, Xiamen 361005, China; 19903723142@163.com
[3] School of Economics and Management, Tsinghua University, Beijing 100084, China; zhangliang@tsinghua.edu.cn
[4] Guanghua School of Management, Peking University, Beijing 100091, China; guanyuqiong@stu.pku.edu.cn
[\*] Correspondence: 18134415371@163.com

**Abstract:** Sustainable development has become a global consensus, and green innovation is the key starting point, and it has become a ballast stone and stabilizer for regional ecological environmental protection and high-quality economic development. Based on GIS tools and multi-method models, this paper studies the spatio-temporal characteristics and influence mechanism of green innovation in three major urban agglomerations in China from 2010 to 2019 from the perspective of economic geography. The study found that: (1) the green innovation efficiency of the three major urban agglomerations in China is in a fluctuating upward trend as a whole, with obvious spatial disequilibrium; (2) from the spatial point of view, the characteristics of global spatial agglomeration distribution have positive spatial correlation, and the law of local spatial autocorrelation is obvious, and the spatio-temporal pattern transitions from "low efficiency, big difference" to "high efficiency, small difference" during the study period; (3) there are obvious spatial spillover and diffusion effects on the green innovation efficiency of the three major urban agglomerations in China as a whole. However, the spatial dependence of green innovation efficiency is inconsistent in China's three major urban agglomerations; the YRD Urban region and the PRD Urban region show a positive impact, while the JJJ Urban region shows a negative impact; (4) the level of economic development, the operating environment of science and technology, and the guiding factors of government system function with significant differences and regional spatial heterogeneity on the efficiency of green innovation in the three major urban agglomerations in China.

**Keywords:** green innovation; three major urban agglomerations in China; spatio-temporal evolution; spatial effect; influencing factors

## 1. Introduction

### 1.1. Background

The report of the 19th CPC National Congress clearly pointed out that we (the party and state people) should unswervingly implement the five new development concepts of "green, innovation, coordination, openness, and sharing"[1]. The construction of China's three major urban agglomerations should adhere to the high-quality development of

economy and ecology as the starting point and foothold. Green development led by innovation is the only way to achieve high-quality economic development; green innovation arises at this historic moment [2]. As a special regional urban form, urban agglomeration has become an important platform to support national economic growth and participate in international competition and cooperation, but it is also facing serious environmental problems [3]. Since the reform and opening up, China's economy has continued to grow at a high speed, but the regional gap is also widening. How to narrow the gap and promote the collaborative relationship between regional economic development and environmental protection has become a hot issue of social concern. The goal of improving the efficiency of regional green innovation is not only in line with the concept of green development, but also encourages the implementation of innovation-driven development strategy. At the same time, it is also the inevitable choice to pursue the green, high-quality, and sustainable development of the regional economy.

As the focus of the main functional regionalization and optimizing the development area, urban agglomeration is the main direction of regional coordinated development. Under the promotion of the new economic normal and the new urbanization development strategy, the construction and development of urban agglomeration has been gradually promoted to an important strategic position. Among them, as the important carriers of China's five regional development strategies, the YRD, JJJ, and PRD urban agglomerations play an important role in China's economic and social development [4]. At the same time, these three urban agglomerations are also important driving forces for China's high-quality sustainable development of the future. In 2019, the combined urban agglomeration of JJJ, YRD, and PRD accounted for only 5.18% of the country's land area, for 24.29% of the country's total population, and contributed 44% of the GDP. The have become the "three major engines" of China's economic development and the "giant" leading the country's economic development. However, it must be recognized that with rapid economic development, environmental problems faced by the three major urban agglomerations in China are extremely serious. There is a large amount of energy consumption and environmental pollutant emissions. In 2018, the total energy consumption of urban agglomerations in the YRD, JJJ, and PRD urban agglomerations accounted for 32.9% of the country's total energy consumption. Industrial wastewater discharge is 4.443 million tons, industrial sulfur dioxide discharge is 0.7773 million tons, and industrial smoke (dust) discharge is 1.0715 million tons, which is still at a high level. According to the 2018 Global Environmental Performance Index, China's environmental performance index ranks 120th among the world's 180 economies, and the air-quality index ranks 177th. More than 70 per cent of China's cities have long been plagued by haze, and the ambient air quality is not up to standard, reflecting the negative effects of rapid economic growth on the environment [5]. On 22 September 2020, General Secretary Xi Jinping announced in his speech at the 75th General Debate of the United Nations General Assembly that China will increase its autonomous national contribution, adopt stronger policies and measures, strive to peak $CO_2$ emissions by 2030, and strive to achieve carbon neutrality by 2060. It shows that China adheres to the strategy of green and low-carbon development and its determination to actively deal with global climate change and environmental protection. Green innovation has not only become an effective way to promote China's green and low-carbon sustainable development, but also a new engine of China's economic development [6]. Therefore, the research on the efficiency of urban green innovation in China's three major urban agglomerations has important reference value and international significance for both China and the world.

Green innovation has received widespread attention in recent years and has become a mainstream concept in environmental management. In terms of research perspectives: scholars have explored green innovation from different perspectives, such as green city perspective [7,8], regional perspective [9], resource capacity perspective [10], innovation value chain perspective [11], sustainable environmental benefits perspective [12–15], corporate, strategic perspective [16–18], and innovation economics perspective [19]. In terms

of efficiency dimension measurement: scholars have conducted in-depth analysis of green innovation efficiency, such as measurement of green innovation performance [20,21], characteristics of performance [22], environmental regulation [23–27], manufacturing [28,29], foreign direct investment [30], green finance [31,32], green technology innovation system [33–35], carbon trading system [36], spatially linked networks [37–39], and digital development [40]. As an effective means to promote the renewal of the mode of economic development and green development, green innovation has solved the problems of new engine, transformation, and upgrading that the country urgently needs to achieve high-quality economic development. However, the impact and risks caused by some "innovation failure" effects on the economy need to be paid attention. Therefore, bringing the factor of "innovation failure" into the green innovation research framework will fundamentally support us to achieve the goal of high-quality and sustainable economic development [32]. With the opening of trade and the introduction of foreign investment, energy consumption is increasing, environmental pollution is becoming more and more serious, and the task of pollution control is arduous. The factor of "environmental pollution" has become the key to restricting the green and high-quality development of the region under the increasingly stringent environmental regulation. What is the impact of "innovation failure" and "environmental pollution" on the development of green innovation efficiency in China's three major urban agglomerations? These problems need to be discussed.

Based on the existing studies, most scholars have done a lot of research on regional or industrial green innovation and its influencing factors. However, there are also the following problems to be studied. First, there are few studies on the efficiency of regional green innovation by bringing "innovation failure" and "environmental pollution" into the research framework. Most of the existing literature analyze the efficiency of regional green innovation from the perspective of "innovation success", while ignoring the influence of "innovation failure". Second, some of the existing literature regard the research area as a completely homogeneous closed system, ignoring the impact of geographical and spatial connections on the efficiency of regional green innovation to some extent. Third, some the existing studies take the provincial scale as the basic unit, which weakens the coping strategies for the core space of multi-scale and multi-level green innovation to a certain extent.

The innovations of this study are mainly reflected in the following points. First, the input and output indicators of green innovation activities are redefined, and the factors of "innovation failure" and "environmental pollution" are included in the non-expected output. The SBM-DEA model and the kernel density model are constructed to measure the efficiency of green innovation and analyze the spatial non-equilibrium of the three major urban clusters in China. Second, the spatial effects are incorporated into the research framework, and the spatial autocorrelation of regional green innovation efficiency is analyzed by using the global and local spatial models to explore the spatial heterogeneity of green innovation efficiency development in China's three major urban clusters. Third, in order to improve green innovation performance and promote the balanced development of China's three major urban agglomerations, we refine the study unit to 48 prefecture-level cities in China's three major urban agglomerations. From the perspective of economic geography, a fixed-effects spatial econometric panel model is integrated to explore the influencing factors and spatial spillover effects of green innovation efficiency.

The contributions of this study are mainly reflected in that, theoretically, this paper expands the research perspective of green innovation, extends the analysis of the spatio-temporal characteristics of incremental green innovation on the basis of the existing total green innovation, and systematically analyzes the multiple drivers of green innovation, which helps to deeply understand the spatio-temporal evolution pattern of green innovation and its influence mechanism in three major urban clusters in China. In practice, by portraying the characteristics of green innovation in China's three major urban agglomerations and using spatial analysis techniques of GIS and spatial measurement tools such as ESDA, this paper reveals the spatial and temporal evolution patterns and driving mechanisms of green innovation in China's three major urban agglomerations and attempts to

propose differentiated policy recommendations for the development of green innovation in China, with a view to providing references for green innovation decision-making in China and other countries and regions with similar conditions.

The rest of the paper is organized as follows. Section 2 first details the development status and functional positioning of the three major urban agglomerations, namely, JJJ, YRD, and PRD, and then gives the research methodology, selection criteria of input–output indicators and data sources, respectively. Section 3 gives the results of green efficiency measurement and spatial autocorrelation analysis of the three major urban agglomerations in China. On this basis, the mechanisms of influencing factors and spatial effects of green innovation efficiency in China's three major urban agglomerations are further explored. Section 4 we discuss the results in more detail from the perspectives of academics and policy-makers. Section 5 we summarize the conclusions and analyze the research limitations and remedial measures of the paper, and, finally, propose future research directions.

### 1.2. Literature Review

The related research on green and innovation at home and abroad is, basically, based on the following three main lines for expansion analysis. The first is to explore the connotation and essence of green innovation and to carry out green innovation system research; the second is to use two kinds of models represented by DEA and SFA to measure the efficiency of green innovation and analyze the spatial pattern of efficiency evolution; the third is to examine the influence mechanism of green innovation efficiency from the perspectives of environmental regulation, foreign investment, shadow economy, technological innovation, and industrial structure, and explore the ways to improve the efficiency of green innovation.

### 1.2.1. Research on the Connotation of Green Innovation

Green innovation is the product of combining traditional innovation theory with the concept of green development on the basis of novelty and value characteristics, carrying the innovative behavior of resource conservation and environmental protection [41]. The "double externality" is a typical feature of green innovation, i.e., the coexistence of positive externalities of innovation outcomes and positive externalities of environmental benefits, which leads to market failure and government failure, and also indicates that there is a unique evolutionary logic of green innovation in terms of technological conditions and innovation investment [42]. Blättel-Mink first put forward the concept of green innovation, emphasizing the ecological dimensions that enterprises take into account in production, operation, market development, and other related strategies [43].

Foreign empirical studies have focused on green innovation behaviors at the industry and firm levels [44,45], explored the characterization indicators and evaluation methods of green innovation capabilities, and studied the paths of green innovation enhancement in terms of environmental policy tools, R&D expenditures, and human capital [46]. Early domestic research on green and innovation focused on green technology innovation, exploring the adoption of green innovation technologies that conserve resources and reduce environmental pollution to achieve sustainable economic development [47–49]. Since then, green technology innovation has evolved toward a broader focus on green innovation that harmonizes people and nature.

### 1.2.2. Research Related to Green Innovation Efficiency Measurement

Most of the existing studies measuring green innovation efficiency follow the traditional stochastic frontier analysis (SFA) and data envelopment analysis (DEA) methods. Xiao et al. [50] measured the GIE in China (2001–2015) with the help of an improved stochastic frontier model (SFA). Yang et al. [51] took 26 cities in China as the research subjects and used the DEA model to measure the GIE of each city from 2010 to 2017. Li et al. concluded that the traditional DEA model did not take into account the factor "slack" when

measuring green innovation efficiency, and used the DEA-SBM model to measure the traditional green development efficiency, green innovation development efficiency, and green total factor productivity in different regions of China (2001–2017), and compared this with the DEA CCR model to reach a more scientific conclusion of SBM [52]. With the deepening of research, the proposal of the super-efficient DEA model not only solves the problem that traditional DEA ignores unexpected output, but also can refine the efficiency difference of effective units, which has gradually become the mainstream method to measure the efficiency of green innovation. For example, Peng et al. used the super-efficient SBM model and Malmquist index to analyze the green technology innovation efficiency of science and technology SMEs in Hebei Province from both static and dynamic aspects, respectively [53]. Li et al. [54] took pollution-intensive industries as an example, incorporated energy input and environmental pollution into the accounting framework of industrial innovation efficiency, and constructed the SBM directional distance function and Luenberger index.

### 1.2.3. Studies Related to Spatial Differences in Green Innovation Efficiency

Liu [55] deeply analyzed the spatial distribution characteristics of the GIE of the regional innovation system and tested its convergence by using an efficiency evaluation method and ESDA method. Different factors such as regional development level and resource endowment determine the differential characteristics of green innovation efficiency in spatial distribution [56]. There are also scholars who have conducted a lot of fruitful studies on their spatial differences based on different scales. For example, Qian et al. found that the level of green technology innovation in the eastern region of China is higher than that in the central and western regions after studying the regional differences in green technology innovation efficiency of industrial enterprises in China, and the gap continues to expand [57]. Peng et al. studied the green innovation efficiency of the Yangtze River Economic Belt and found that the differentiated characteristics of green innovation efficiency levels in the downstream, midstream, and upstream were obvious [58]. Overall, the studies at either scale can prove that there is an uneven and insufficient growth of green innovation efficiency among regions in China.

### 1.2.4. Studies Related to the Influencing Factors of Green Innovation Efficiency

Green innovation efficiency is a typical indicator that takes into account both economic and ecological characteristics, and, thus, is influenced by a variety of factors, and it is generally believed that the level of economic development can bring more innovation factor inputs, thus stimulating enterprises to accelerate innovation and efficient innovation [59]. Environmental regulations can lead to the occurrence of "pollution paradise", but also force enterprises to strengthen the research and development of pollution control technologies and equipment, stimulating the efficiency of green innovation [60,61]. Thus, the impact of environmental regulation has diversified characteristics [62]. FDI brings not only capital but also advanced technology and management experience, therefore, the technology spillover effect of FDI can be used to enhance China's green innovation capability. Ji et al. [63] proposed that FDI has a significant threshold effect on technological innovation, that there are significant regional differences in the spillover effect of FDI on technological innovation, and that unreasonable intensity of environmental regulations, lower level of economic development and human capital are the main constraints on the positive technological spillover effect of FDI and environmental regulations in some regions of China. Therefore, FDI brings environmental pollution while making up for China's capital shortage. In addition, other scholars have examined the effects on green innovation efficiency from carbon-trading policies [64], environmental rights trading [36], industrial structure upgrading [65], green finance [66], fiscal decentralization [67], high-tech industrial agglomeration [68], low-carbon city construction [69], and institutional systems [70,71], but the differences in research regions and research methods have led to different findings.

*1.3. Purpose and Questions*

In summary, academics have conducted fruitful research on green innovation, but there is still much room for expansion. First, the research scale is focused on macroscopic areas, such as national and provincial areas, and there are few micro and mesoscopic studies on urban clusters. In particular, the green innovation efficiency of China's three major urban agglomerations is still in the blank stage, which leads to insufficient demonstration and guidance of research results. Secondly, it emphasizes the examination of green innovation efficiency influencing factors from the perspective of the region as a whole, ignoring that the heterogeneity of economic development levels and location endowments of different regions may lead to different degrees of spatial spillover of green innovation efficiency influencing factors.

In view of this, this study uses the super-efficient SBM model considering unexpected output, kernel density analysis, exploratory spatial analysis (ESDA), and spatial econometric model to focus on the following problems.

(1) What are the temporal changes and spatio-temporal evolution of green innovation efficiency in China's three major urban agglomerations as a whole? (2) What are the spatial clustering and distribution characteristics of green innovation efficiency in each city of the three major urban agglomerations in China? (3) What is the mechanism of the influence of each driver on green innovation efficiency, and is there any spatial spillover effect? (4) How to propose differentiated green innovation policies for the effective investment of green innovation resources based on the above research findings?

## 2. Research and Design

*2.1. Study Area*

In recent years, with the rapid economic growth, the connection between regions will become closer, especially between regions with geographical connection. It is understood that the current Chinese urban agglomerations mentioned by academia and the government currently involve 19 major (Beijing–Tianjin–Hebei, Yangtze River Delta, Pearl River Delta, Shandong Peninsula, Central Liaoning, Middle Reaches of Yangtze River, Central Plains, Chengdu–Chongqing, Guanzhong Plain, West Coast of the Straits, Harbin and Changchun, Central and Southern Liaoning, Huobao–Erhu–Yulin, Ningxia Along the Yellow, Lanzhou Regions, Central Shanxi, Central Guizhou, Central Yunnan, and Beibu Gulf urban agglomerations). However, most of these urban agglomerations are still in the initial stage of development, and the relevant research results are not many and do not fully meet the definition of urban agglomerations in the strict sense. In contrast, geographically, the JJJ, YRD, and PRD urban agglomerations are located in the developed regions of North, East, and South, respectively, in China. As the main channels for China to connect with the world economy, they are the most mature, international, and competitive urban agglomerations in China, with strong regional representativeness.

These three urban agglomerations have a total area of about 400,000 km² and cover the political, economic, cultural, and financial centers of the country and first-tier cities such as Beijing, Shanghai, Guangzhou, and Shenzhen, contributing more than 40% of the GDP in 2019. These three major urban agglomerations represent the highest form and direction of urban cluster development in China [72]. At the same time, the Chinese government clearly proposed that the JJJ, YRD, and PRD urban agglomerations should accelerate the formation of new advantages in international competition, participate in international cooperation and competition at a higher level, and play an important supporting and leading role in national economic and social development [3]. Therefore, this paper takes JJJ, YRD, and PRD urban agglomerations as the research objects, and develops a study on the spatial and temporal characteristics of green innovation efficiency and its influencing factors of these three major urban agglomerations, and the research results have strong demonstration and guiding effects. The vector map data of China's three major urban ag-

glomerations are drawn based on relevant data from the China Basic Geographic Information Center, as shown in Figure 1. (http://ngcc.sbsm.gov.cn, accessed on 10 January 2021).

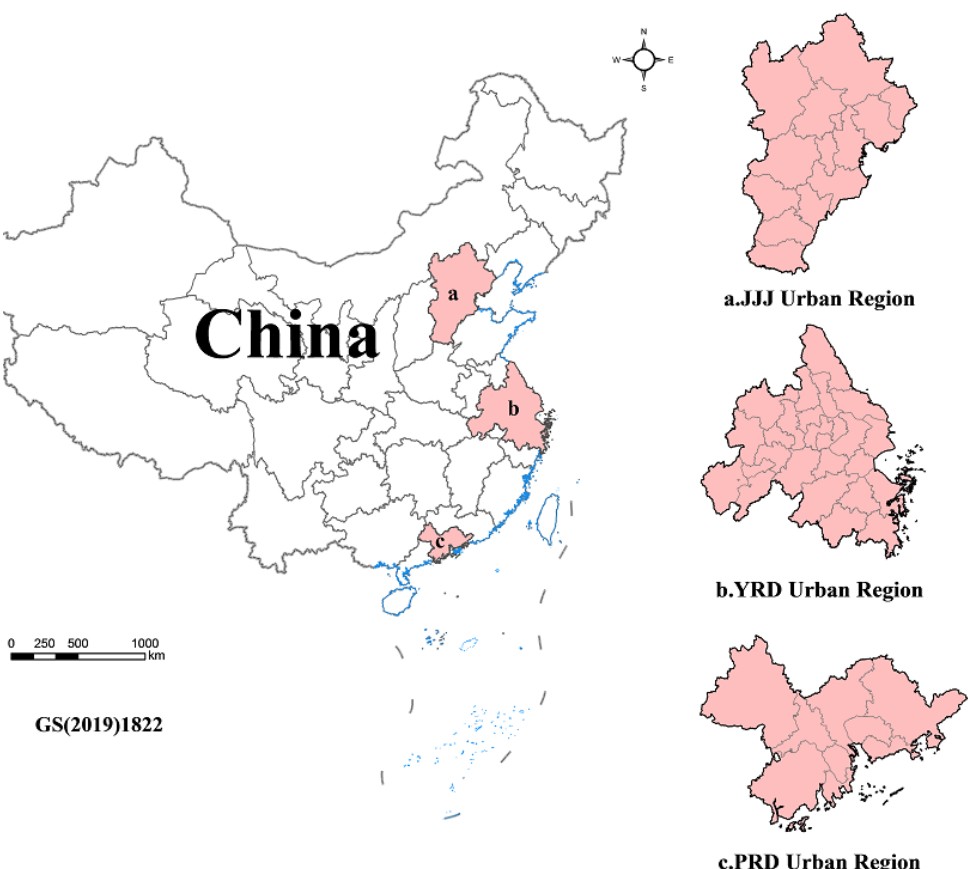

**Figure 1.** Three major urban agglomerations in China (**a**–**c**).

*2.2. Research Methods*

2.2.1. SBM-DEA Efficiency Measurement Model

Most of the traditional DEA models are radial and angular measurements, which cannot fully take into account the relaxation of input and output, and cannot accurately measure the efficiency value when there is unexpected output [52]. In order to overcome these shortcomings, Tone proposed a non-radial and non-angular SBM-DEA model based on relaxation variables. It is assumed that there are n decision-making units in a production system, and each decision-making unit can get $S_1$ expected output and $S_2$ non-expected output by using m inputs. The input vector is expressed as $x \in R^m$ and the expected output vector is expressed as $y^g \in R^{s_1}$. The non-expected output vector is expressed as $y^b \in R^{s_2}$. Definition matrix $XY^g$, $Y^b$ as $X = \left[ x_1, \cdots x_m \right]$, $Y^g = \left[ y_1^g, \cdots, y_{s_1}^g \right]$, $Y^b = \left[ y_1^b, \cdots, y_{s_2}^b \right]$. Among them, $x_i > 0, y_i^g > 0, y_i^b > 0$. The set of production possibilities is: $P = \left\{ (x, y^g, y^b) \,|\, x \geq X\delta, y^g \leq Y^g\delta, y^b \geq Y^b\delta, \delta \geq 0 \right\}$.

According to the SBM model in which Tone contains unexpected output, the model is established as shown in Formula (1). Among them, $\overline{s}^-, \overline{s}^{g+}, \overline{s}^{b-}$ redundant values that represent input, expected output, and non-expected output, $\rho$ represents the total efficiency of the evaluated decision-making unit $(x_0, y_0^g, y_0^b)$, and $0 \leq \rho \leq 1$. If $0 \leq \rho \leq 1$, the evaluated decision-making unit is invalid. If $\rho = 1$, the evaluated decision-making unit is valid.

$$\rho = \min \frac{1 - \frac{1}{m}\sum_{i=1}^{m}\frac{\overline{s_i^-}}{x_{i0}}}{1 + \frac{1}{s_1 + s_2}\left(\sum_{r=1}^{s_1}\frac{\overline{s_r^{g+}}}{y_{ro}^g} + \sum_{k=1}^{s_2}\frac{\overline{s_k^{b-}}}{y_{ko}^b}\right)}, \quad s.t. \begin{cases} x_0 = X\delta + \overline{s}^- \\ y_0^g = Y^g\delta - \overline{s}^{g+} \\ y_0^b = Y^b\delta + \overline{s}^{b-} \\ \overline{s}^- \geq 0, \overline{s}^{g+} \geq 0, \overline{s}^{b-} \geq 0, \delta \geq 0 \end{cases} \tag{1}$$

Kernel density estimation is a nonparametric method for studying spatial disequilibrium distribution, which describes the distribution form of random variables by estimating their probability densities [41]. The kernel density estimation formula is as follows:

$$f_n(x) = \frac{1}{n h_n}\sum_{i=1}^{n} K\left(\frac{x_i - x^-}{h_n}\right) \tag{2}$$

In the form, $f_n(x)$ for kernel density estimation; $K\left(\frac{x_i - x^-}{h_n}\right)$ is a kernel function; $n$ represent number of samples, $h_n$ represent broadband, and $x^-$ is the mean value.

## 2.2.2. Exploratory Spatial Data Analysis (ESDA)

Examining the spatio-temporal characteristics and spatial agglomeration effects of regional green innovation efficiency first requires testing for the existence of spatial autocorrelation. Exploratory spatial data analysis identifies spatial correlation patterns, spatial regime differences, and other forms of spatial instability of elements by determining their spatial locations. ESDA includes two tools, the first of which is the global Moran's I index, which is used to verify spatial patterns across the study area and describes the overall variability characteristics of a parameter mean across the study area. Its expression is:

$$I = n\sum_{i=1}^{n}\sum_{j=1}^{n}\left(x_i - x^-\right)\left(x_j - x^-\right) \Big/ \left\{\sum_{j=1}^{n}\left(x_i - x^-\right)^2 \sum_{i=1}^{n}\sum_{j=1}^{n} w_{ij}\right\} \tag{3}$$

In Equation (3): $I$ is the global Moran's I index, which takes values in the range [−1, 1]. The more the value of this index tends to 1, the stronger the correlation in the green innovation efficiency space. The value of this index tends to 0, which indicates that the green innovation efficiencies of the three major urban agglomerations in China are independent of each other and follow a random distribution in space. $X_i, X_j$ represents the green innovation efficiency measures of cities i and j, respectively; $x^-$ is the arithmetic mean of green innovation efficiency of all cities; n is the number of cities studied; $w_{ij}$ is the adjacency weight matrix, which indicates the adjacency relationship between two cities, and $w_{ij}$ = 1 when i and j are adjacent, otherwise it is 0.

The second tool is the local spatial autocorrelation index, which is usually measured by using the local Moran's I index and plotting the LISA map agglomeration, measuring the spatial correlation characteristics of the region and the neighboring regions, including high–high agglomeration, low–low agglomeration, high–low outlier, and low–high outlier. The calculation formula is:

$$I = n\left(x_i - x^-\right)\sum_{j=1}^{n} w_{ij}\left(x_j - \overline{x}\right)^2 \Big/ \sum_{i=1}^{n}\left(x_i - x^-\right) \tag{4}$$

In Equation (4): $X_i, X_j$, $w_{ij}$ and other symbols are defined as in (3), $I$ is local Moran's I indices, taking values in the range [−1, 1]; positive values of indices indicate spatial clustering of similarity (high or low) around regional units, and negative values of indices indicate spatial clustering of non-similarity (high or low) around regional units.

### 2.2.3. Spatial Econometric Model

On the basis of testing the spatial autocorrelation of green innovation efficiency in three major urban agglomerations in China, a spatial econometric model is further constructed. Compared with traditional regression methods, the spatial econometric model takes into account the spatial relevance and spatial dependence of complex samples [73]. Therefore, this paper uses the spatial econometric model to decompose the main factors affecting the green innovation efficiency of the three major urban agglomerations in China, and study their spatial spillover effects. Common models include (SEM), (SAR), and (SDM).

### 2.3. Index Selection

Combined with the existing literature, the input–output index of regional innovation activities is redefined, in which the innovation input index includes R&D full-time personnel equivalent and R&D capital stock, the innovation expected output index includes the number of green invention patent applications and new product sales income, and the unexpected output includes industrial wastewater emissions, industrial waste gas emissions, and bank non-performing loan year-on-year ratio. The setting and data processing of each index are described as follows: For the input of innovation activities, two aspects are characterized in terms of R&D personnel input and financial input. In terms of R&D personnel input, in order to better measure the amount of human input and actual working hours of R&D personnel in innovation activities, the indicator of full-time equivalent of R&D personnel in each region is selected for measurement. In terms of capital input, considering the influence of the accumulation of prior investment on innovation output, the perpetual inventory method is used to account for the R&D capital stock of each region, the equation is: $K_{it} = (1-\delta) \times K_{i(t-1)} + R_{i(t-1)}$, where $K_{it}$, $K_{i(t-1)}$ denotes the R&D capital stock of region i in periods t and t − 1, respectively, $\delta$ denotes the depreciation rate of the R&D capital stock, which is set to 15%, $R_{i(t-1)}$ denotes the actual internal expenditure of R&D funding in region i in period t − 1, and is obtained by dividing the nominal expenditure obtained by dividing the R&D price index, which is calculated using 0.85* consumer price index + 0.15* fixed-asset investment price index [74], and the formula $K_{i0}$ for the R&D capital stock in each region in the base period is: $K_{i0} = R_{i0}/(g+\delta)$, g is the growth rate of internal expenditure on R&D funding. As a result, the R&D capital stock of each region with 2010 as the base period was calculated.

The expected output of green innovation activities is considered in terms of knowledge technology output and product output and ecological environment. From the perspective of knowledge technology output, considering that among the three types of patents—invention, utility model, and design—invention patents have higher technology content and better reflect the original innovation capability of the region, and are less restricted by the patent examination and licensing agencies, the number of invention patent applications granted is selected as an indicator to characterize the knowledge technology output of innovation. However, the general number of patent applications and the number of patents granted cannot truly reflect "green" innovation. Therefore, this paper draws on Yu Peng et al.'s green patent classification method, and uses "green recycling, green innovation" as keywords to obtain patent data of the three major urban clusters in China from 2010 to 2019 on the patent search and analysis service platform of the State Intellectual Property Office (SIPO), and classifies the applicants' regions in order to the quantity and quality of green innovation invention patents are measured comprehensively. For the perspective of product output, reflecting the innovation achievements of a region from the dimension of transformation of scientific and technological achievements, new-product sales revenue is a good indicator to measure [30]. Given that the statistical caliber of the China Science and Technology Statistical Yearbook has been changed from "large and medium-sized industrial enterprises" to "industrial enterprises above scale" for new-product sales revenue since 2011, the new-product sales revenue of large and medium-

sized industrial enterprises in each region (2010) and the new-product sales revenue of industrial enterprises above the scale (2011–2019) is the expected output. It should be noted that since the measurements of regional green innovation efficiency in this paper are all cross-sectional comparisons within the same year, no statistical caliber adjustment and price deflations are required for this indicator. In terms of ecological and environmental benefits, considering the problems of soil erosion and desertification caused by the sloppy use of resources, the expected output of green technology innovation is characterized by the greening coverage of built-up areas, which can better reflect the ecological value of green innovation activities [72]. The unexpected output of innovation activities is considered in terms of two factors: "innovation failure" and "environmental pollution". Regarding the "innovation failure" factor, Schumpeter (1912) argues that innovation is the creation of a new production function to obtain potential excess profit, which clearly reflects that the purpose of innovation is to obtain excess profit. If economic profit is obtained, the innovation is successful; if not, the innovation is a failure. Therefore, the success or failure of innovation is marked by whether or not economic profit is obtained. On the other hand, in order to relieve financial pressure, enterprises may apply for additional loans from commercial banks, which may result in non-performing loans if they cannot be compensated by profits. Therefore, the year-on-year ratio of non-performing loans from commercial banks is used to characterize "innovation failure". For the "environmental pollution" factor, the existing literature usually uses the volume of "three waste" pollutants (i.e., industrial wastewater, exhaust gas, and solid waste) to represent unexpected output, but considering that the vast majority of China's industrial solid waste generation has been disposed of and used in recent years (according to the China Statistical Yearbook (2017), in 2016, China's general industrial solid-waste disposal utilization rate has reached 80.73%, dumping and discarding only 0.01% of the generated amount), the amount of dumping and disposal has decreased significantly. Therefore, the study selects industrial wastewater and industrial waste gas emissions to characterize "environmental pollution".

The data are obtained from the 2011–2020 China Statistical Yearbook, the China Urban Statistical Yearbook, the China Environmental Statistical Yearbook, the China Financial Yearbook, the China Science and Technology Statistical Yearbook, the China Energy Statistical Yearbook, and panel data from 48 prefecture-level cities' national economic and social development bulletins, environmental bulletins, etc. For the missing data of individual years, the interpolation method is used to supplement. Given the complexity of the process of green innovation activities, a certain time interval is required for input–output transformation. According to the assumptions of previous studies [31], the delay time from input to output is assumed to be one year in this paper. Therefore, both input indicators and environmental variables in this paper are treated with a one-period lag (Table 1).

**Table 1.** Input–output indicators of green innovation efficiency.

| Indicator Type | Indicator Composition | Indicator Representation | Unit |
|---|---|---|---|
| Input index | Manpower input | R&D practitioner full-time equivalent | people/year |
| | Capital investment | R&D capital stock | million yuan |
| | Energy input | Electricity consumption of the whole society | million KW.h |
| | | Total water supply | ten thousand tons |
| Output index | Economic output | New product sales revenue | million yuan |
| | Technical output | Number of green invention patent applications granted | million |
| | Ecological output | Greening coverage of built-up areas | % |
| | Non-expected output | Year-on-year ratio of commercial banks' non-performing loan amounts | % |
| | | Industrial wastewater discharge | million tons |

| | |
|---|---|
| Industrial waste gas emissions | million standard cubic meters |

## 3. Result Analysis

### 3.1. Analysis of the Measurement Results of Green Innovation Efficiency

This paper uses MaxDEA software to measure the green innovation efficiency of three major urban agglomerations in China from 2010 to 2019, and shows it with Origin software according to the calculation results (see Figure 2).

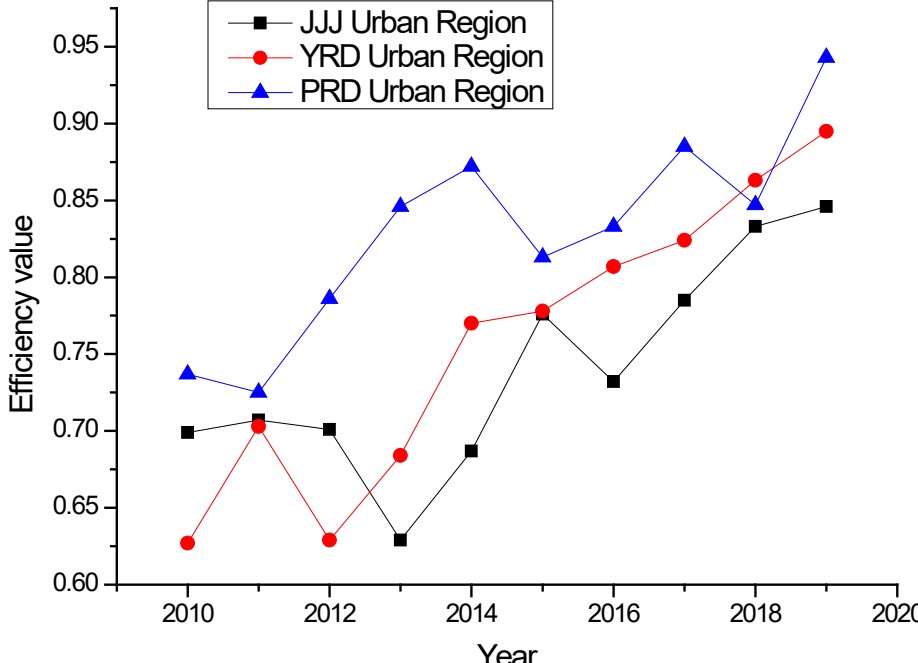

**Figure 2.** Calculation results of green innovation efficiency of three major urban agglomerations in China from 2010 to 2019.

According to Figure 2, overall, the average green innovation efficiency of China's three major urban agglomerations increased from 0.667 in 2010 to 0.863 in 2019, with a comprehensive increase of 29.39%, but there is still plenty of room for improvement. Among them, the JJJ Urban region rose from 0.699 in 2010 to 0.846 in 2019, an increase of 21%; the YRD Urban region rose from 0.627 in 2010 to 0.895 in 2019, an increase of 42.74%; and the PRD Urban region rose from 0.737 in 2010 to 0.943 in 2019, an increase of 27.95%, showing a development trend of "YRD Urban region > PRD Urban region > JJJ Urban region".

With the passage of time, the green innovation efficiency of China's three major urban agglomerations shows a fluctuating upward trend as a whole. During the study period, the green innovation efficiency of China's three major urban agglomerations showed a flat "N" trend as a whole. This is closely related to China's environmental policy on green innovation and development. Since 2013, China has successively introduced the "12th five-year Plan for the Development of Green Manufacturing Science and Technology", "several measures to promote Scientific and technological Innovation", and the five new development concepts of "green, innovation, coordination, openness, and sharing" put forward by the 19th CPC National Congress, as the representative of the policy-oriented drive [36]. The government has accelerated supply-side structural reform, optimized the allocation of innovative resources, and continued to increase investment in innovative capital and human capital, and the level of green innovation in China has ushered in a window period for development.

### 3.2. Spatial Disequilibrium Analysis of Green Innovation Efficiency

(1)  Kernel density estimation analysis

According to the Gaussian kernel density estimation method, the kernel density estimation distribution map of green innovation efficiency of the three major urban agglomerations in China from 2010 to 2019 is drawn by Matlab (see Figure 3). From 2010 to 2019, the nuclear density distribution curve shifts to the right as a whole, indicating that the green innovation efficiency of the three major urban agglomerations in China is gradually increasing. From a morphological point of view, the core density curve of green innovation efficiency shows a typical bimodal distribution, and there is an obvious gap between the main peak and the secondary peak, indicating that there is a certain degree of classification of green innovation efficiency in China's three major urban agglomerations. That is, there is obvious spatial disequilibrium. From the kurtosis point of view, the fluctuation of the main peak height of the nuclear density curve increases, and the efficiency change interval has a decreasing trend, indicating that the regional differences of green innovation efficiency among the three major urban agglomerations in China have decreased.

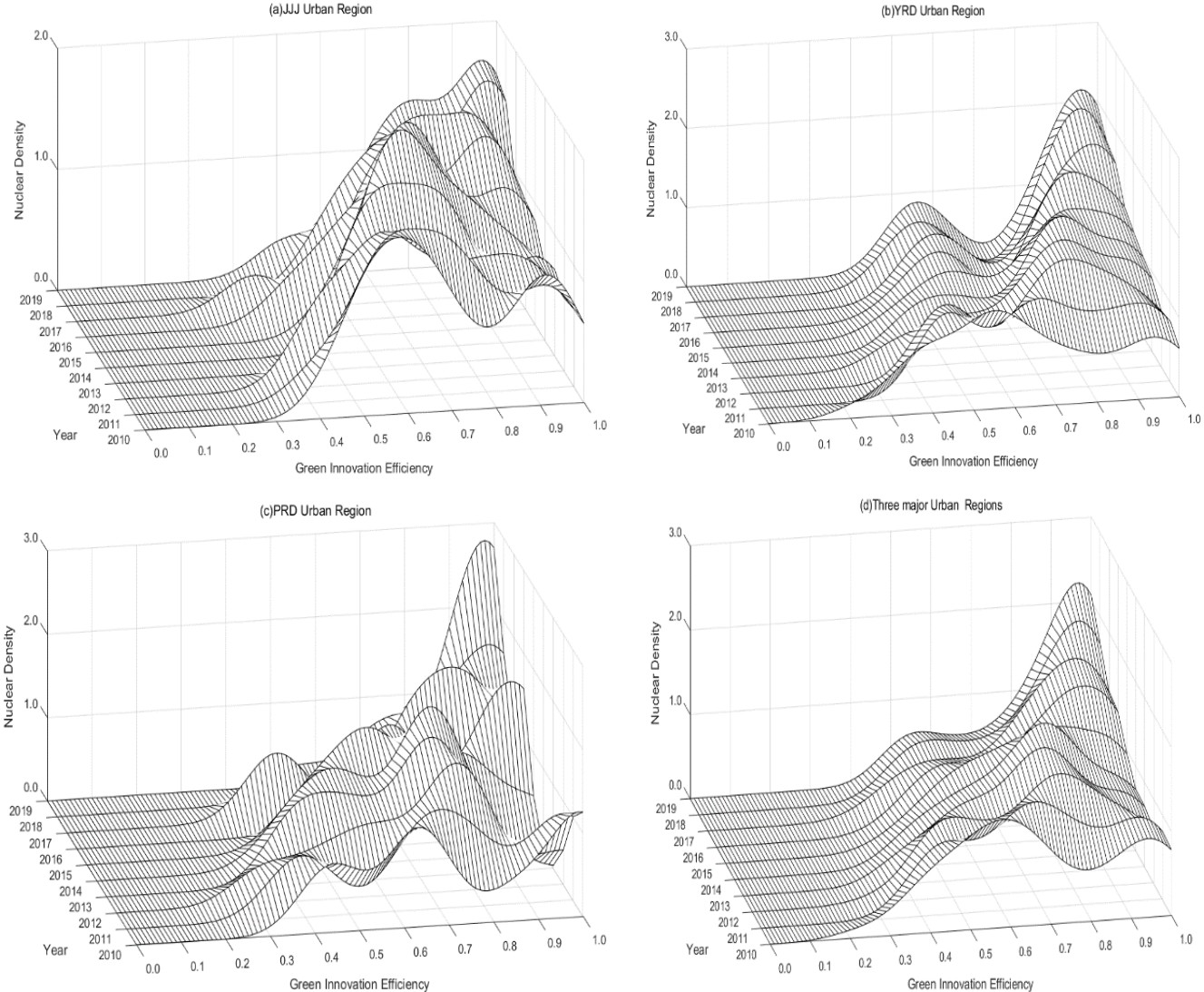

**Figure 3.** Results of nuclear density analysis: (**a**) JJJ Urban Region, (**b**) YRD Urban Region, (**c**) PRD Urban Region, and (**d**) three major Urban agglomerations.

(2)  Analysis of the characteristics of spatio-temporal evolution

In order to further analyze the spatial differences of green innovation efficiency in China's three major urban agglomerations, the spatial and temporal differences of green innovation efficiency in 48 cities of China's three major urban agglomerations in 2013, 2016, and 2019 were drawn by using Arcgis 10.2 software. With regard to the division of efficiency levels, the study draws lessons from the practices of previous scholars and combines the measurement results of green innovation efficiency of the three major urban agglomerations in China [37–39], to demarcate four efficiency zones of high, medium and high, medium, and low efficiency: high-efficiency zone (0.900–1.000); medium- and high-efficiency zone (0.751–0.899), medium-efficiency zone (0.651–0.750), and low-efficiency zone (0.000–0.650). Details of the spatio-temporal pattern differentiation map are shown (see Figures 4–6).

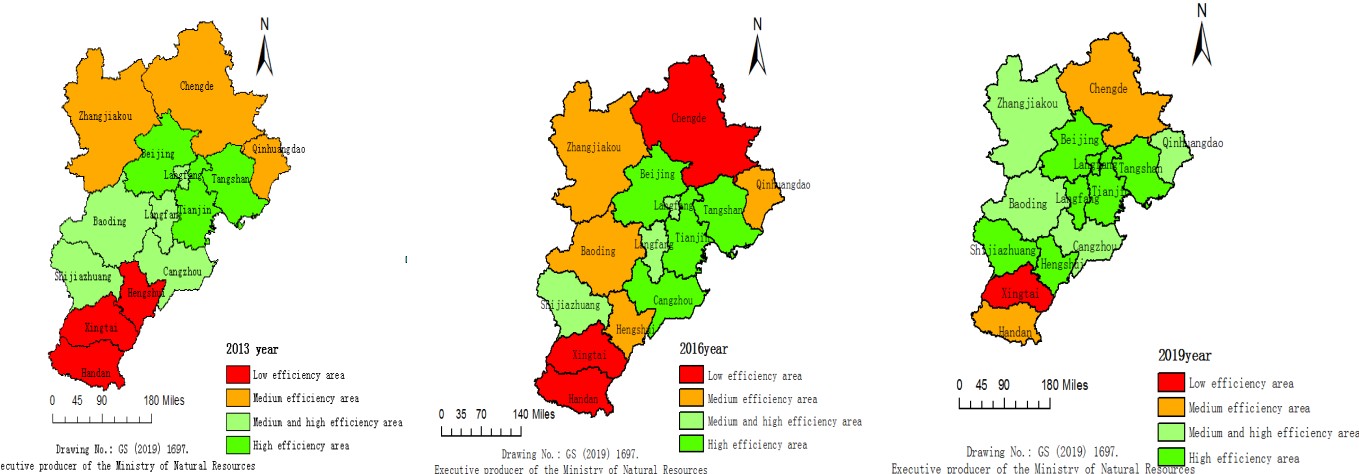

**Figure 4.** Spatial pattern map of green innovation efficiency of JJJ Urban region.

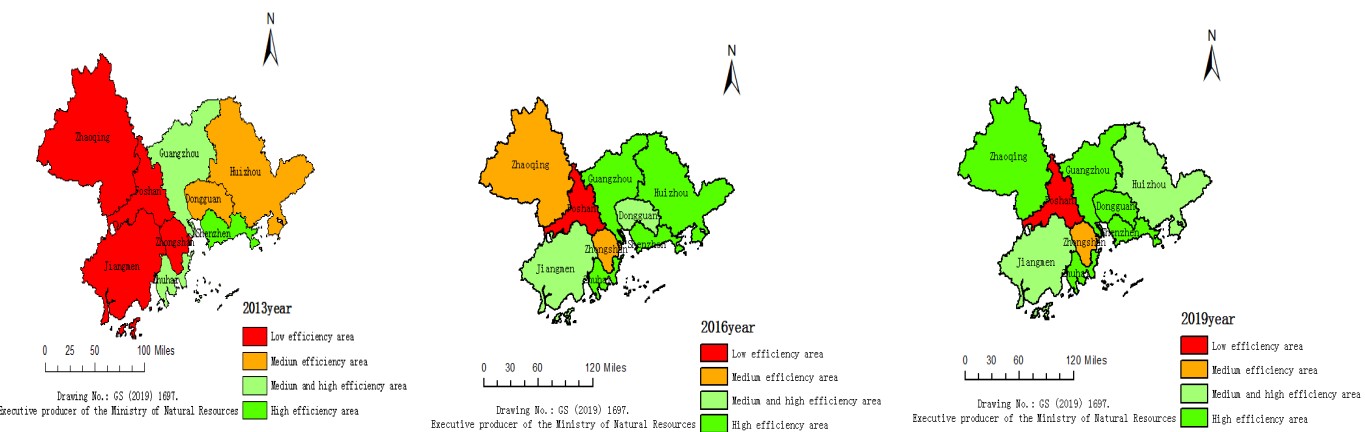

**Figure 5.** Spatial pattern map of green innovation efficiency of the PRD Urban region.

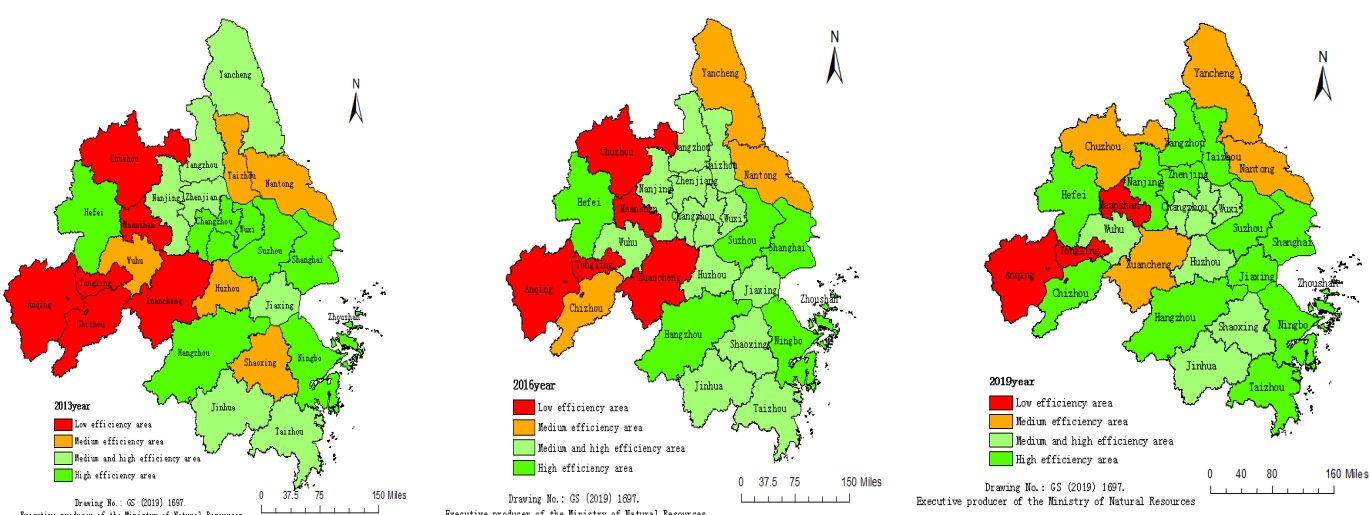

**Figure 6.** Spatial pattern map of green innovation efficiency of the YRD Urban region.

According to the results of Figures 4–6, it can be seen that, at the overall level, the green innovation efficiency of China's three major urban agglomerations has shown an increasing temporal and spatial pattern evolution trend from 2010 to 2019, which indicates that in the past decade, it has actively responded to the green innovation development strategy proposed by the state, and achieved remarkable results. Specifically: (1) JJJ Urban Region: from 2010 to 2019, the number of cities in high-efficiency zones increased by 23.08%, and the number of cities in inefficient areas decreased by 30.77%, indicating that the JJJ Urban agglomerations actively promoted the integrated and coordinated development strategy. Specific to the city level, the innovation efficiency of green in Beijing, Tianjin, and Tangshan has been in the high-efficiency zone during the inspection period, Langfang City has gradually transitioned from the middle–high-efficiency zone to the high-efficiency zone, Xingtai City has been in the inefficient zone, and the remaining cities have continuously improved their green innovation efficiency over time. Xingtai City's urban economic development is relatively backward, the competitiveness of talent innovation is insufficient, and the development of the heavy-duty and extensive industrial structure has led to a lack of vitality of green innovation elements. Xingtai City should rely on the JJJ Urban Region integrated coordinated development strategy and actively introduce advanced talents, technologies, and management experience to improve its own level of green innovation and reduce the emission of "three wastes". (2) PRD Urban Region: during the inspection period, the number of cities in high-efficiency zones increased by 46.7%, and the number of cities in inefficient areas decreased by 33.3%, which reflects the positive response of the PRD Urban Region to the guidance and forcing of green, low-carbon, and innovative environmental policies. In recent years, a series of green reform documents, such as the "Green and Low-carbon Development of the PRD Urban Region 2020 Vision and Goals", have been introduced, which have played a good guiding and constructive role and significantly improved the complementarity of industries and resources between cities centered on Guangzhou and Shenzhen, and seen the transformation and upgrading of production technology and the coordinated treatment of pollution emissions. Also, the joint responsibility of innovation costs has promoted the rapid improvement of green innovation efficiency. (3) YRD Urban Region: cities in high-efficiency areas increased by 25%, and in low-efficiency areas decreased by 14.3%. On the whole, it shows the space–time pattern of the east is high and that of the west is low. In the coastal area it is higher than in the inland area, and there are the geographical characteristics of the "core–edge", which basically forms the development trend spread, with Shanghai, Hangzhou, Nanjing, and Hefei as the center.

In order to test whether this spatial clustering phenomenon occurs randomly or there is a specific distribution law, it is necessary to further explore the spatial distribution law of green innovation efficiency.

### 3.3. Exploratory Spatial Data Analysis (ESDA)

(1) Global autocorrelation analysis

According to Formula (3), combined with the measured value of green innovation efficiency, using adjacency distance as spatial evaluation weight, the spatial autocorrelation of green innovation efficiency in three major urban agglomerations in China is tested and analyzed, and the global Moran's I index test results are calculated by using Stata software (see Table 2). The results show that the global Moran's I of the three major urban agglomerations in China from 2010 to 2019 is positive and has passed the statistical test of 5%. It shows that the spatial distribution of green innovation efficiency of the three major urban agglomerations in China is not random, but shows obvious positive spatial correlation, and the cities with high (or low) green innovation efficiency are often adjacent. From the perspective of the overall Moran's I evolution trend, it roughly shows an upward trend of "*N*" fluctuation, rising from 0.016 in 2010 to 0.038 in 2014, then to 0.015 in 2016, and then to 0.029 in 2019. Indicating that with the evolution of time, the spatial correlation of green innovation efficiency in China's three major urban agglomerations gradually increases in the fluctuation.

**Table 2.** Global Moran's I of green innovation efficiency of the three major urban agglomerations in China from 2010 to 2019.

| Year | JJJ Urban Region | | PRD Urban Region | | YRD Urban Region | | Three Major Urban Regions | |
|---|---|---|---|---|---|---|---|---|
| | Moran's I | Z | Moran's I | Z | Moran's I | Z | Moran's I | Z |
| 2010 | 0.025 *** | 2.457 | −0.043 *** | 3.646 | 0.032 *** | 2.875 | 0.016 *** | 1.978 |
| 2011 | 0.026 *** | 2.561 | 0.049 *** | 4.138 | 0.023 *** | 2.328 | 0.024 *** | 2.347 |
| 2012 | 0.017 ** | 1.987 | 0.058 ** | 4.497 | 0.035 * | 3.237 | 0.037 ** | 3.517 |
| 2013 | 0.043 *** | 3.665 | 0.051 *** | 4.208 | 0.046 *** | 3.809 | 0.028 *** | 2.659 |
| 2014 | 0.068 *** | 5.337 | −0.055 *** | 4.388 | 0.053 ** | 4.334 | 0.038 *** | 3.434 |
| 2015 | 0.042 *** | 3.599 | 0.057 *** | 4.443 | 0.056 *** | 4.417 | 0.036 *** | 3.217 |
| 2016 | 0.013 ** | 1.871 | 0.062 *** | 4.659 | 0.058 * | 4.497 | 0.015 ** | 1.968 |
| 2017 | 0.018 * | 2.137 | 0.063 *** | 4.724 | 0.061 ** | 4.582 | 0.023 ** | 2.332 |
| 2018 | 0.034 *** | 3.051 | 0.071 *** | 5.345 | 0.077 ** | 5.836 | 0.036 *** | 3.337 |
| 2019 | 0.048 *** | 3.987 | −0.076 *** | 5.743 | 0.082 *** | 6.019 | 0.029 *** | 2.783 |

Note: *, **, *** indicate significant at the statistical level of 1%, 5%, and 10%, respectively.

Table 2 reveals the overall spatial autocorrelation characteristics of green innovation efficiency in the three major urban agglomerations in China. In order to further explore the local spatial relationships, it is necessary to combine the local autocorrelation index measurement and draw LISA atlas mapping to reveal the local spatial heterogeneity of the three major urban agglomerations in China.

(2) Local autocorrelation analysis

According to Equation (4), the green innovation efficiency measures of 48 prefecture-level cities in the three major urban agglomerations in China in 2013, 2016, and 2019 were selected, and LISA agglomeration maps were obtained using ArcGIS and GeoDa software. The spatial agglomeration types of green innovation efficiency in JJJ (Figure 7), YRD (Figure 8), and PRD (Figure 9) urban agglomerations are classified as "HH-High Efficiency", "LH-Hollow", "LL-Low Efficiency" and "HL-Polarized" to measure the agglomeration status and hot and cold distribution pattern of each region.

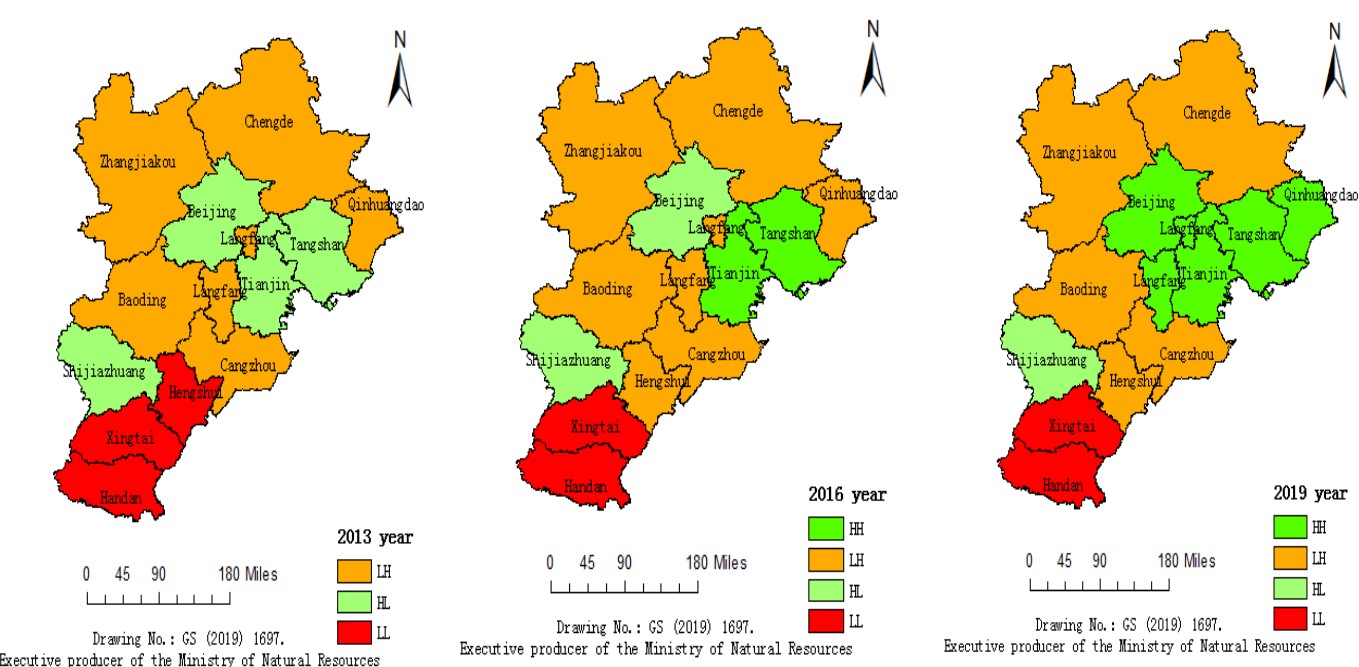

**Figure 7.** LISA agglomeration map of green innovation efficiency of JJJ Urban region.

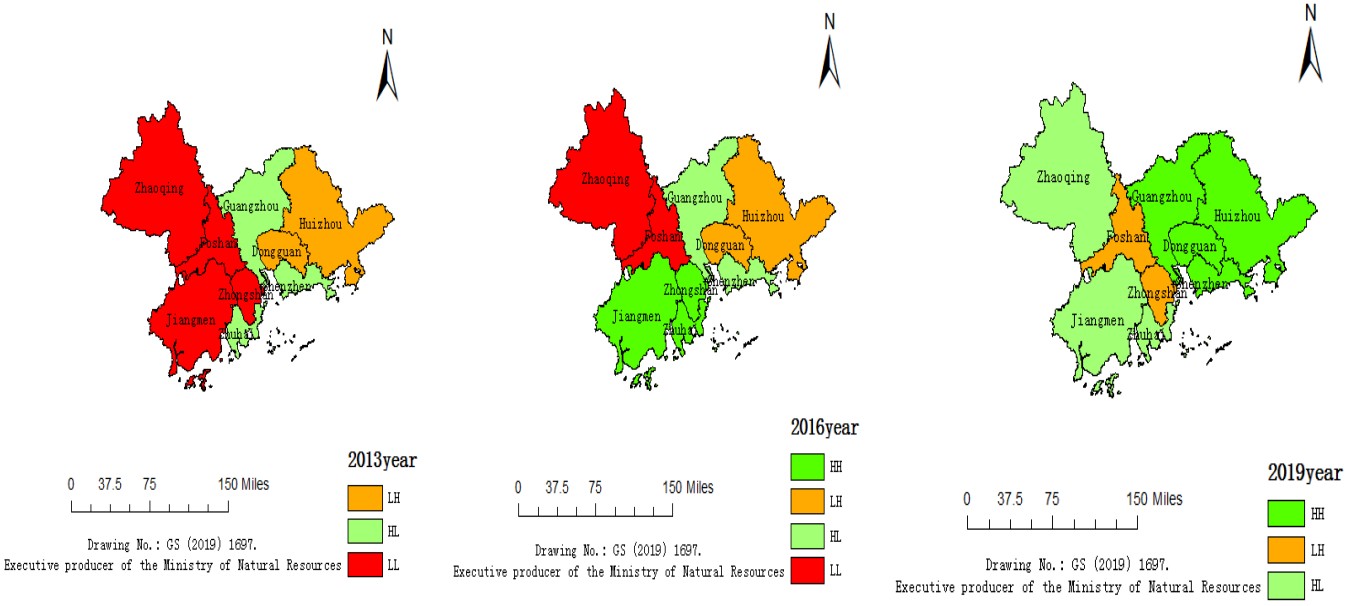

**Figure 8.** LISA agglomeration map of green innovation efficiency of PRD Urban region.

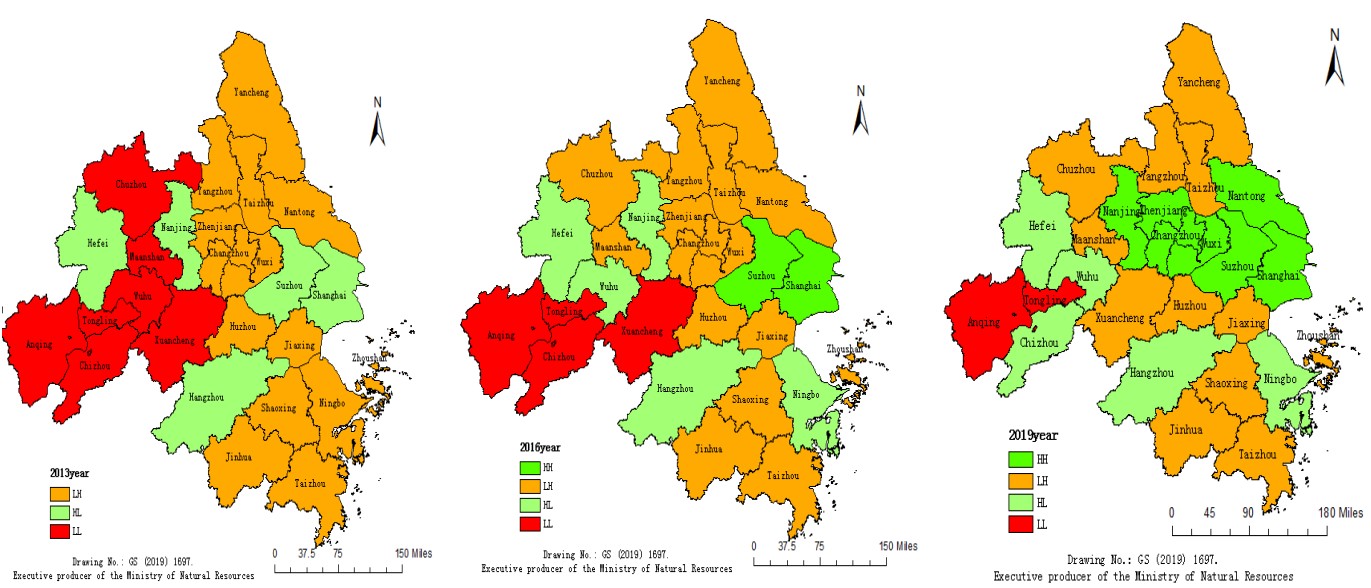

**Figure 9.** LISA agglomeration map of green innovation efficiency of YRD Urban region.

From the perspective of local autocorrelation spatio-temporal evolution pattern, the local spatial autocorrelation regularity is obvious, although different agglomeration areas show a certain expansion and contraction in the spatio-temporal evolution. However, during the study period, the agglomeration types of the three major urban agglomerations in China showed a spatio-temporal transition from "low-efficiency big differences" to "high-efficiency small differences". Specifically: ① JJJ Urban region: from 2010 to 2019, "HH-high efficiency type" cities increased by 38.46%, while "LL-low efficiency type" cities decreased by 7.69%. As a whole, it presents the distribution of time and space pattern with Beijing, Tianjin, Tangshan, and Shijiazhuang as the core. It shows that the strategic effect of actively promoting integrated and coordinated development in JJJ Urban region is obvious. ② PRD Urban region: from 2010 to 2019, "HH-high efficiency type" cities increased by 44.44%, while "LL-low efficiency type" cities decreased by 44.44%, indicating that the PRD Urban region has made remarkable progress during the study period and achieved remarkable results. It is necessary to maintain a good momentum in the future. Especially driven by the radiation of Guangzhou and Shenzhen, the regions should give full play to the multiplier effect and form the growth pole of regional development. ③ YRD Urban region: from 2010 to 2019, "HH-high efficiency type" cities increased by 26.92%, while "LL-low efficiency type" cities decreased by 19.23%, showing a positive upward trend as a whole, basically forming a development trend of spreading around with Shanghai, Suzhou, Hangzhou, Nanjing, and Hefei as the center.

### 3.4. Spatial Regression Analysis of Influencing Factors

On the basis of testing the existence of certain geospatial dependence of green innovation in the three major urban agglomerations in China, this paper uses a spatial econometric model to decompose the main factors of green innovation efficiency in order to better reveal the existence of spatial effects, since it is difficult to incorporate spatial characteristics in traditional econometric models [42].

(1)  Selection of influencing factors

The development of green innovation is a complex system affected by multi-agents and multi-factors, and its influencing factors come not only from the innovation subject, but also from the innovation environment factors inside and outside the system. This paper combines the existing research results [59–71], as well as the imbalance of economic development, openness, human capital, and ecological environment of the three major

urban agglomerations in China. This study is carried out from three aspects: the level of economic development, the operating environment of science and technology, and the guidance of government system. This summarizes eight major influencing factors and explores the dynamic mechanism and optimization direction of improving the efficiency of green innovation in China's three major urban agglomerations. Among them, the level of economic development is the basis of green innovation, the operating environment of science and technology is the driving force of green innovation, and the orientation of government system is the guarantee of green innovation.

The level of economic development considers both the average output of the economy and the structure of economic output. The level of per capita income and industrial structure are selected to characterize it. First, the green innovation process is rooted in a geographical environment with different economic background conditions. The introduction of innovation agents, innovation output, and the transformation and application of innovation results are closely related to the economic output conditions on which the innovation process depends [66]. Second, the industrial structure is closely related to the undesired outputs in the green innovation system and is an important factor in achieving sustainable development of green innovation [11].

The operating environment of science and technology should consider both domestic and international aspects. Select the level of financial development, the level of opening to the outside world, and the level of urban informatization to characterize it. First, improving the financial system not only provides financial support for enterprise innovation and R&D activities, but also disperses the corresponding risks for investors and promotes the rational allocation of innovation resources in the market [3]. Second, the level of urban informatization is conducive to the flow of green innovative talents and resource elements inside and outside the region. It may promote the accumulation of elements in the region and produce innovation spillover effects on the surrounding areas. It may also lead to the flow of innovative elements to areas with good development environment and high administrative efficiency, forming a siphon effect [46]. Third, foreign investment can make up for the shortage of local funds, facilitate the flow of innovative elements, bring advanced management models and marketing experience, and then promote the efficiency of green innovation. But at the same time, it will also bring competitive effect, resulting in low-end locking [63].

Government institutional orientation, considering both input and output aspects, is chosen to characterize the level of education and environmental regulation. First, technological progress is an important driving force for the development of green innovation, and education is a necessary path to promote technological progress [75]. The cultivation of scientific and technological talents and the stimulation of scientific and technological innovation power cannot be achieved without the power of education, and the education system has become an important aspect of government support. Second, the influence mechanism of environmental regulation on green innovation includes "following cost theory" and "Porter hypothesis". In the initial stage of the implementation of environmental policy, it is difficult for low environmental cost to play an incentive role in innovation [26]. "Follow the cost theory" holds that the increased governance costs of environmental regulation will crowd out innovation investment, which is not conducive to the improvement of green innovation efficiency [30]. With further improvement of the intensity of environmental regulation, the rising cost of pollution forces enterprises to implement technological innovation [62]. The Porter hypothesis holds that reasonable external environmental regulation can offset the cost of environmental regulation for a long time, so as to achieve a win–win situation of regional environmental and economic benefits, and help to improve the efficiency of green innovation [61].

Based on the above analysis, considering the availability and accuracy of the data, eight impact indicators of green innovation efficiency at three levels—the level of eco-

nomic development, the operating environment of science and technology, and government regulation—are selected for empirical analysis. All the data are from the China Urban Statistical Yearbook from 2011 to 2020 (see Table 3).

**Table 3.** Definition and explanation of related variables of driving factors of urban green innovation efficiency.

| Influence Level | Influencing Factors | Variable Abbreviation | Measurement Index | Unit |
|---|---|---|---|---|
| Economic development level | Per capita income level | *PGDP* | Per capita GDP | Person/yuan |
| | Industrial structure | *INDU* | Output value of secondary industry/GDP | % |
| | | *SERV* | Output value of tertiary industry/GDP | % |
| Operation environment of science and technology | Financial development level | *FINA* | Balance of deposits and loans of financial institutions | Yuan |
| | The level of opening up | *FDI* | Foreign direct investment/GDP | % |
| | Urban informatization level | *INTERNET* | Total output value of Post and Telecommunications Services/GDP | % |
| Government system orientation | Intensity of environmental regulation | *ER* | Industrial pollution Control Expenditure/GDP | % |
| | Higher education level | *STU* | Number of college students | Person |

(2)  Result analysis

Stata15 software is used to analyze the influencing factors of green innovation efficiency in the three major urban agglomerations in China by spatial panel regression analysis. In order to identify the validity of the spatial econometric model, the following test steps are taken [51,52]. First of all, the adaptability of SAR or SEM model is judged by LM test, the results show that the LM test value of both is significant at 1% level, but the robust LM test value of SAR model cannot pass the 5% significance test, indicating that SEM is more suitable. Secondly, the Hausman test judgment model adopts fixed effect or random effect and the results show that the statistical value is 27.413, the degree of freedom is 8, the accompanying probability is 0.003, and we reject the original hypothesis, so, we choose the fixed-effect model. Finally, LR and Wald tests are carried out to see whether the SDM model can be further reduced to a spatial error model (SEM). If it can be degraded, then adopt a more targeted degraded model, and if it cannot be degraded, then use a more inclusive SDM model, and the result *p*-value shows that it accepts the original hypothesis, that is, it can be degraded. The test results are shown in Table 4.

**Table 4.** Test results of spatial econometric model.

| Inspection | Statistical Value | *p*-Value | Inspection | Statistical Value | *p*-Value |
|---|---|---|---|---|---|
| LM (LAG) | 52.247 *** | 0.000 | LM (ERR) | 46.734 *** | 0.000 |
| R-LM (LAG) | 37.264 | 0.073 | R-LM (ERR) | 12.718 *** | 0.000 |
| Spatial effect-LR | 564.257 *** | 0.000 | Time effect-LR | 105.673 *** | 0.000 |
| Wald spatial lag | 15.208 *** | 0.000 | Wald spatial error | 15.109 *** | 0.000 |
| LR spatial lag | 14.994 *** | 0.001 | LR spatial lag | 14.930 *** | 0.001 |

Note: *** indicate significant at the statistical level of 1%.

Based on the above test results, this paper selects the best fitting spatial and temporal fixed-effect SEM model to analyze the influencing factors of green innovation efficiency of the three major urban agglomerations in China. Set the model to:

$$EFFI_{it} = a_1 PGDP + a_2 SERV + a_3 ER + a_4 INTERNET + a_5 INDU + a_6 FDI + a_7 STU + a_8 FINA + u_{it},$$
$$u_{it} = \rho \sum_{j=1}^{N} W_{it} u_{jt} + \varepsilon_{it} \tag{5}$$

where: $EFFI_{it}$ efficiency for green innovation; $a_1 \sim a_8$ is the coefficient to be estimated; $W_{it}$ is the weight matrix, $\rho$ is a matrix coefficient; $\varepsilon_{it}$ is a random error term; and logarithmic processing is taken to eliminate the heteroscedasticity part of the data.

According to the statistical results (Table 5), the regression coefficients $\rho$ of the three major clusters in China are significantly positive, indicating that there is a substantial spatial spillover and diffusion effect on the green innovation efficiency of the three major urban agglomerations in China as a whole. It shows that this region has a positive effect on the efficiency of green innovation in neighboring areas. However, the spatial dependence of green innovation efficiency is inconsistent in China's three major urban agglomerations. Among them, the YRD Urban region and PRD Urban region has a positive impact on the green innovation efficiency of the neighboring areas, while the JJJ Urban region has a negative impact on the green innovation of the neighboring areas. The reason is that there is a certain competitive relationship among different regions in terms of capital, labor force, energy input, and so on. The distribution of innovation resources in China is uneven, the level of economic development in the YRD Urban region and PRD Urban region is high, the innovation elements are more dynamic, the market-oriented innovation incentive mechanism is relatively perfect, and it is easy to form a positive impact relationship between regions. However, the economic development of JJJ Urban region is relatively backward, innovation resources are relatively scarce, and the well-developed areas have siphon effect, showing a negative spatial spillover effect.

**Table 5.** Statistics of spatial econometric regression results.

| Explanatory Variable | Three Major Urban Agglomerations | JJJ Urban Region | YRD Urban Region | PRD Urban Region |
|---|---|---|---|---|
| ln*INDU* | −0.006 *** | −0.010 *** | −0.004 ** | −0.009 ** |
| | (0.049) | (0.052) | (0.037) | (0.029) |
| ln*SERV* | 0.004 ** | 0.003 * | 0.001 ** | 0.007 *** |
| | (0.002) | (0.002) | (0.001) | (0.002) |
| ln*ER* | 0.002 *** | 0.004 *** | 0.003 ** | 0.001 ** |
| | (0.000) | (0.001) | (0.002) | (0.001) |
| ln*INTERNET* | −0.001 | −0.013 * | 0.007 | 0.039 |
| | (0.002) | (0.014) | (0.002) | (0.021) |
| ln*PGDP* | 0.421 *** | 0.394 *** | 0.546 *** | 0.684 *** |
| | (0.039) | (0.042) | (0.027) | (0.043) |
| ln*FDI* | 0.018 ** | 0.016 *** | −0.013 ** | −0.014 ** |
| | (0.009) | (0.004) | (0.007) | (0.007) |
| ln*STU* | 0.126 *** | 0.075 *** | 0.137 *** | 0.219 *** |
| | (0.003) | (0.000) | (0.007) | (0.002) |
| ln*FINA* | 0.454 | −0.433 | 0.275 | 0.886 |
| | (0.036) | (0.054) | (0.087) | (0.029) |
| $\rho$ | 0.452 *** | −0.242 ** | 0.368 *** | 0.417 *** |
| | (0.003) | (0.015) | (0.007) | (0.000) |
| Sigma_2e | 0.038 *** | 0.052 *** | 0.032 *** | 0.040 *** |
| | (0.002) | (0.004) | (0.002) | (0.003) |
| $R^2$ | 0.32 | 0.27 | 0.24 | 0.35 |

Note: standard error is in parentheses. *, ** and *** indicate significant at 10%, 5% and 1% confidence level, respectively.

The level of economic development. (1) The regression coefficient of per capita income level is positive and passed the significance test of 1%, indicating that it has a positive impact on the efficiency of green innovation. The overall income level of the region has increased, the attraction of innovative talents has increased, the financing capacity of enterprises has been enhanced, and the efficiency of regional innovation has been improved. In terms of sub-regions, the per capita income regression coefficient of urban agglomeration in the PRD Urban region is the highest, because the awareness of green innovation in economically developed areas is relatively strong, and a relatively perfect system has been formed in terms of innovation infrastructure, talent system, and so on. It has stronger innovative resource allocation and output ability. (2) The regression coefficient of the secondary industrial structure is negative, and the regression coefficient of the tertiary industrial structure is positive, and all of them have passed the significance test of 5%, indicating that green innovation in China's three major urban agglomerations is mainly driven by the development of green innovation in the tertiary industries.

The operating environment of science and technology. (1) The regression result of financial development level is not significant, indicating that the financial development level of China's three major urban agglomerations does not have a significant impact on regional green innovation. The reason is that capital and manpower, as the decisive elements of green innovation, need to rely on a sound financing system, but the financial system with commercial banks as the core has the phenomenon of "ownership discrimination" and "scale discrimination". Financial mismatch reduces the ability of financial institutions to share the risk of enterprise innovation activities and the allocation of resources, so it is difficult to play a role in promoting green innovation. (2) The regression coefficient of urban informatization is not significant, indicating that improving the level of urban informatization cannot improve the efficiency of regional green innovation. Although the high level of informatization can reduce the flow cost of innovation factors and improve the level of inter-regional technical cooperation, but it will also form the siphon effect of central cities on talents and funds in the surrounding areas. Especially in Beijing, Shanghai, Shenzhen, Guangzhou, and other areas, there is a phenomenon of "one city dominating", and the excessive agglomeration of innovation elements in the same space, the impact of the two effects on the level of urban informatization is not significant. (3) The regression coefficient of foreign direct investment level is positive and has passed the significance test of 5%, which shows that it plays a role in promoting the efficiency of green innovation as a whole. However, the functional relationship shows significant spatial heterogeneity. The regression coefficient of urban agglomeration in PRD Urban region and YRD Urban region is negative, while that of JJJ Urban region is positive. The reason is that due to the limitations of geographical location and the level of economic development, the foreign investment attraction of JJJ Urban region is relatively low, and improving the level of opening up can effectively promote the efficiency of green innovation. While the PRD Urban region and YRD Urban region has a large intensity of foreign investment, and the over-reliance on foreign capital will hinder regional independent research and development to a certain extent, which is not conducive to the improvement of regional green innovation level.

The orientation of government system. (1) The regression coefficient of the level of higher education is positive and passed the statistical test of 1%, indicating that the development of higher education has a positive effect on the improvement of the level of green innovation. Higher education is the engine of scientific and technological progress, which plays a prominent role in the formation of human capital and the promotion of green innovation. (2) The regression coefficient of environmental regulation is positive and passed the significance test of 1%, indicating that environmental regulation has a positive effect on the efficiency of green innovation. The reason is that with the increasingly prominent resource and environmental problems of China's three major urban agglomerations and the background of China's new development concept, the government formulates environmental protection policies according to its own environmental development problems.

The process of enterprise production and governance continues to be green, and the positive impact of environmental regulation on green innovation is prominent. However, the regression coefficient is relatively small, indicating that the current environmental regulation is in the exploratory stage and needs to be further improved to the "appropriate environmental regulation" of Porter's hypothesis to achieve a win–win situation of coordinated development of economy and environment.

(3) Robustness test

In order to ensure the reliability of the research conclusions, the following reliability tests need to be carried out. The selection and setting of spatial weights can determine the different spatial relationships between variables. Due to the limitation of the assumed relationship of spatial adjacency weight matrix, the spatial correlation effects of geographical distance and economic activities are not taken into account. Therefore, according to the related research [33,38], the spatial weight matrix of geographical distance, the spatial weight matrix of economic distance, and the nested spatial weight of economic geography are constructed to re-estimate the benchmark model.

From the regression results (Table 6), it can be seen that although the absolute values of the regression coefficients and spatial item coefficients of the benchmark model in the three cases are lower than those of the previous article, the significance and symbols are basically consistent with the previous conclusions. This shows that although the measurement method of spatial weight matrix has changed, it also maintains a conclusion similar to the previous empirical results, indicating that the empirical results are robust.

**Table 6.** Regression results of robustness test.

| Explanatory Variable | Geographical Distance | Economic Distance | Economic Geography Nesting |
|---|---|---|---|
| ln*INDU* | −0.003 *** | −0.001 *** | −0.004 ** |
| | (0.009) | (0.082) | (0.137) |
| ln*SERV* | 0.002 ** | 0.000 * | 0.001 ** |
| | (0.000) | (0.023) | (0.017) |
| ln*ER* | 0.001 *** | 0.001 *** | 0.001 ** |
| | (0.000) | (0.001) | (0.032) |
| ln*INTERNET* | −0.001 | −0.000 | 0.002 |
| | (0.012) | (0.014) | (0.002) |
| ln*PGDP* | 0.362 *** | 0.394 *** | 0.286 *** |
| | (0.000) | (0.002) | (0.007) |
| ln*FDI* | 0.013 ** | 0.009 *** | −0.013 ** |
| | (0.019) | (0.000) | (0.027) |
| ln*STU* | 0.098 *** | 0.075 *** | 0.037 *** |
| | (0.000) | (0.000) | (0.000) |
| ln*FINA* | 0.329 | −0.332 | 0.278 |
| | (0.016) | (0.053) | (0.074) |
| $\rho$ | 0.376 *** | −0.442 ** | 0.428 *** |
| | (0.000) | (0.005) | (0.017) |
| Sigma_2e | 0.023 *** | 0.035 *** | 0.032 *** |
| | (0.014) | (0.007) | (0.000) |
| $R^2$ | 0.29 | 0.27 | 0.24 |

Note: standard error is in parentheses. *, ** and *** indicate significant at 10%, 5% and 1% confidence level, respectively.

## 4. Discussion

### 4.1. Theoretical Value

Comparing the results of this study with those of previous studies, it is not difficult to find some similarities and differences. First, the results of this paper indicate that the regional green innovation development in the three major urban agglomerations in China shows spatial heterogeneity and a spatio-temporal distribution pattern of positive spatial clustering, which is basically consistent with the findings of Yuan, Mi, Liu, Huang, and Xiao [3,4,11,32,50]. In addition, Corradini said that green technology patents are very unevenly distributed among European countries [76]. Kijek and Matras-Bolibok pointed out that countries with high and medium–high eco-innovation capacity are in northern and central-western Europe, while those with medium–low and low eco-innovation capacity are in central-eastern and southern Europe [77]. It can be seen that the spatial cluster and spatial differences exhibited by green innovation are geographic phenomena common to different regions and different countries. Secondly, this paper found that the three level driving variables of economic development level, science and technology operating environment and government institutional orientation have a positive impact on green innovation in three major urban clusters in China. This is consistent with some findings of Zhao, Wang, Fan, Zhang, and others [31,35,61,62]. However, Wu and Feng et al. also noted that environmental regulation and R&D inputs have a dampening effect in some regions of China [26,30], but this was not found in this study. In addition, Saunia et al. identified economic and institutional pressures as the main drivers of green innovation in Finnish equine companies [78]. Han et al. stated that innovation capacity and environmental regulations promote eco-innovation in Myanmar SMEs [79]. However, Cuerva et al. argue that innovation inputs, such as R&D capital and human capital, promote traditional innovation but not green innovation in Spanish SMEs [80]. Brunnermeier et al. also stated that increased environmental regulation was not effective in stimulating environmental innovation in the U.S. manufacturing sector because firms feared that regulators would raise regulatory standards when developing new technologies [29]. As can be seen, the results of this paper are broadly consistent with those of Yuan, Mi, Liu, Huang, Xiao, Corradini, Kijek, and others, but not fully consistent with those of Cuerva, Brunnermeier, Wu, Feng, and others. The conclusions may be due to differences in research methods, country context, case themes, and thresholds of the indicators themselves, leading to differences. In addition, this paper also explores the role of education level on green innovation, which complements the existing index system of influencing factors. Existing studies pay less attention to the spatio-temporal clustering characteristics and influence mechanisms of regional green innovation, while this paper pays more attention to them, which helps to re-examine regional green innovation from an incremental perspective. The analysis results show that the spatio-temporal characteristics of green innovation in the three major urban agglomerations in China have certain similarities, but there are differences in the intensity of their driving factors and influence mechanisms.

In particular, this paper also finds differences in the driving forces of eight green innovation efficiency impact indicator variables for the three major urban agglomerations in China, indicating that the driving mechanisms of green innovation vary across regions, which may provide references for the optimization of green innovation policies in different regions. For example, we find that FDI investment has an inhibitory effect on green innovation in the Yangtze River Delta and Pearl River Delta city clusters, further suggesting that when receiving foreign investment, the YRD and PRD urban agglomerations should adhere to the concept of ecological and environmental protection and set entry thresholds for highly polluting and energy-consuming projects to prevent the transfer of pollution to the city clusters. This is basically consistent with the findings of Kuang, Ji et al. [59,63]. In addition, it is worth noting that our study found that the share of secondary industry output in GDP in the industrial structure became a key factor limiting the development of green innovation in the three major urban agglomerations, suggesting that the

three major urban agglomerations in China need to accelerate industrial green transformation and upgrading, and use the characteristics of industrial restructuring and local resource endowments to find an economic model suitable for sustainable regional development. This is basically consistent with the findings of Du et al. [65]. In addition, it was also found that the level of per capita income and the level of higher education have a relatively large role in promoting the development of green innovation in the three major urban agglomerations in China and are crucial to the development of regional green innovation, which echoes the findings of Mi, Ji, Wang, et al. [4,63,75].

### 4.2. Policy Enlightenment

According to the matrix of Boston Consulting Group (BCG), drawing on the ideas of the model [81], it corresponds exactly to the four spatial agglomeration types of the three major urban agglomerations in China: "H-H, L-L, H-L, and L-H". The government should comprehensively consider the spatial characteristics of different cities as well as regional development conditions, formulate differentiated green innovation policies, effectively and precisely allocate green innovation resources, formulate and adopt strategies and countermeasures in line with its own reality, which will help narrow the gap of green innovation development between regions and promote coordinated and sustainable regional economic development. H-H: this type of zone should continue to maintain good development momentum, give full play to scientific and technological resources, technology accumulation, human capital, regional policies and other advantages, and increase investment in R&D. At the same time, it should also play the role of technology diffusion and radiation drive, and shoulder the heavy responsibility of driving the green and coordinated development of the region. L-L: the state should further increase the support for this type of zone, give full play to the guiding role and leverage effect of fixed-assets investment, and this type of zone should also focus on the digestion and absorption of the introduced technology, actively learn the advanced experience and technology of the H-H agglomeration zone, dock industrial gradient transfer. And use of scientific and technological innovation to transform the economic development model, to achieve a bending overtake. H-L: this type of zone should drive the surrounding areas to improve the level of green innovation as soon as possible, and actively carry out cross-regional cooperation. L-H: it should actively strive for the positive radiation influence of the H-H agglomeration zone, and enhance the soft and hard strength by introducing experience, strengthening cross-regional cooperation.

Pay great attention to the spatial correlation and uneven characteristics of green innovation activities, and give full play to the spatial spillover effect. Build a cross-regional green innovation cooperation platform and establish a sharing mechanism of innovation resources. Optimize the spatial layout of green innovation activities, give full play to the comparative advantages of each region, and avoid industrial homogenization and vicious competition in the same region. The green innovation growth pole is built, and through the dominant effect, multiplier effect, and spillover and diffusion effect, the green innovation activities of the neighboring cities are radiated and driven, and the "siphon effect" is turned into "radiation effect". By improving the agglomeration effect, scale effect, and ecological effect of the urban agglomerations, it will actively drive the neighboring lagging cities and improve their catch-up effect, and eventually improve the overall development level of the three major urban agglomerations.

Effectively transform government functions and give full play to the guiding role of the government in promoting green technological innovation in enterprises. The government should first create a fair competitive innovation environment and institutional guarantee, insist on market regulation as the main means, and reduce excessive administrative intervention and monopoly in the market. Secondly, the government should support more private and small and micro enterprises that are in urgent need of capital and have strong green innovation capability in terms of fiscal and tax policies, and guide banks to

increase credit support for such enterprises to improve the financing system for innovative subjects. Finally, the government should tend to provide enterprises with appropriate technical and financial support for environmental management, and establish an environmental economic policy system with "inherent restraining power". It should form a long-term mechanism for enterprises to allocate environmental resources effectively, and stimulate them to carry out green technological innovation by establishing a mechanism for transforming green innovation results and strengthening the protection of intellectual property rights of green technologies.

## 5. Conclusions

Based on the panel data of 48 prefecture-level cities of three major urban agglomerations in China from 2010 to 2019. This paper constructs a SBM-DEA efficiency model to measure the green innovation efficiency of each urban agglomeration, and establishes a spatial model to explore the spatio-temporal evolution trend and spatial effect characteristics of green innovation efficiency of the three major urban agglomerations in China. The results show that:

Overall, the average green innovation efficiency of China's three major urban agglomerations increased from 0.667 in 2010 to 0.863 in 2019, with a comprehensive increase of 29.39%, but there is still much room for improvement. In terms of time trend, the green innovation efficiency of the three major urban agglomerations in China shows a fluctuating upward trend as a whole, and the average efficiency is 0.740, 0.758, and 0.829, respectively. As a whole, it presents the spatial distribution pattern of "PRD Urban region > YRD Urban region > JJJ Urban region", which indicates that there are great spatial differences in the development of green innovation efficiency among the three major urban agglomerations in China.

In the aspect of spatial disequilibrium analysis: in the nuclear density analysis, the nuclear density distribution curve shifts to the right over time, indicating that the green innovation efficiency of the three major urban agglomerations in China shows a gradual upward trend, and has obvious spatial disequilibrium. From the kurtosis point of view, the fluctuation of the main peak height of the nuclear density curve increases, and the efficiency change interval has a decreasing trend, indicating that the regional differences of green innovation efficiency among the three major urban agglomerations in China have decreased. The characteristics of the evolution and distribution of spatio-temporal pattern show that during the study period, the cities in the high-efficiency areas of JJJ Urban region increased by 23.08%, the cities in low-efficiency areas decreased by 30.77%. The cities in the high-efficiency areas of PRD Urban region increased by 46.7%, while the number of cities in low-efficiency areas decreased by 33.3%. The cities in the high-efficiency areas of the YRD Urban region increased by 25%, while the cities in the low-efficiency areas decreased by 14.3%, and the spatial agglomeration characteristics of the three major urban agglomerations are relatively obvious.

Spatial analysis: the overall Moran's I of the three major urban agglomerations in China from 2010 to 2019 is positive, passing the statistical test of 5%. It shows that there is a significant spatial correlation between the green innovation efficiency of the three major urban agglomerations in China. From the perspective of local autocorrelation spatio-temporal evolution pattern, the local spatial autocorrelation regularity is obvious. On the whole, the agglomeration types of the three major urban agglomerations in China show a spatio-temporal pattern transition from "low efficiency, big difference" to "high efficiency, small difference". The results of spatial econometric regression show that there are obvious spatial spillover and diffusion effects on the green innovation efficiency of the three major urban agglomerations in China as a whole. It shows that this region has an impact on the efficiency of green innovation in neighboring areas. However, the spatial dependence of green innovation efficiency is inconsistent among the three major urban

agglomerations in China. YRD Urban region and PRD Urban region have a positive impact on green innovation efficiency in neighboring areas. JJJ Urban region has a negative impact on green innovation efficiency in neighboring areas.

The spatial heterogeneity and impact mechanisms of green innovation found in this paper are also present in other developed and developing countries such as Europe [77], Finland [78], and Myanmar [79]. In the context that sustainable development has become a global consensus, both the UK, US, Australia, Japan, Sweden, Belgium, and developing countries such as India, Iran, Malaysia, Vietnam, Turkey, and Egypt are generally under pressure to transition to sustainable development. The research methodology and findings of this paper can also provide a reference for decision making to optimize the development of green innovation in these countries.

However, as a quantitative study, there are certain limitations and future research directions in this paper. For example, this paper is mainly studied at the regional level of the three major urban agglomerations in China, and lacks comparisons at the international perspective level, which leads to some limitations in the applicability of the research results. To further remedy the limitations, we provide a comparative analysis of foreign studies in the discussion section. In this paper, representative internal and external influencing factors (per capita income level, industrial structure, financial development, openness to the outside world, city informatization level, environmental regulation, and higher education level) are selected. The mechanisms of other influencing factors and government green innovation policies on the efficiency of green innovation in cities need to be further studied. Moreover, the empirical research in this paper is mainly based on statistical data, with less attention to corporate subjects and innovation individuals. Therefore, the accuracy of some research findings needs to be further verified. In the next study, we will continue to deepen our research on these three aspects and carry out dynamic simulation and validation of government green innovation policies.

**Author Contributions:** Conceptualization, B.H.; data curation, K.Y.; formal analysis, Y.G. and L.Z.; funding acquisition, B.H.; investigation, K.Y. and T.N.; methodology, K.Y.; resources, B.H.; supervision, B.H.; validation, T.N.; visualization, K.Y.; writing—original draft, Y.G.; writing—review and editing, B.H., L.Z., T.N., and Y.G. All authors have read and agreed to the published version of the manuscript.

**Funding:** National Key R&D Program of China (No. 2019YFC1908302).

**Institutional Review Board Statement:** Not applicable.

**Informed Consent Statement:** Not applicable.

**Data Availability Statement:** Not applicable.

**Acknowledgments:** We thank the editors and anonymous reviewers for their careful reading of our manuscript and their many constructive comments and suggestions.

**Conflicts of Interest:** The authors declare no conflict of interest.

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
