# Peer review of "Study on the Spatial and Temporal Evolution Patterns of Green Innovation Efficiency and Driving Factors in Three Major Urban Agglomerations in China—Based on the Perspective of Economic Geography"

_sustainability, doi:10.3390/su14159239_

Round 1
Reviewer 1 Report
Analysis on Spatial -temporal characteristics and Influencing Factors of Green Innovation Efficiency in Three Major Urban Agglomerations in China.
This paper explains the trends of green innovation efficiency and its influencing factors using spatial analysis. The topic is interesting and very relevant. I have a few comments and suggestions.
1. Just a few writing issues
For example the opening sentence "Under the background of the report of the 19th CPC National Congress clearly 40 pointed out that we should unswervingly implement the five new development concepts" . Who is being referred by "we"? Is this a quote or ?
There are some small mistakes in the paper e.g using capital letters instead of small letters and in the variable definition "Renewd employees" ?
2. The authors need to improve their in-text citation style. For example, in the second paragraph, the authors just put a block of references (3-19) . The authors need to insert the cites next to the related explanation, this makes it easier for the readers.
3. There is a need to add other references on environmental regulation and green innovation. For example, the authors write "Foreign empirical research mainly focuses on the green innovation behavior at the industry and enterprise level..." but there is no reference to a foreign research article on green innovation. They can add Mbanyele et al., (2022) as another reference. They can supplement the three references below with others.
Reference suggestion
Mbanyele, W. and Wang, F., 2022. Environmental regulation and technological innovation: Evidence from China. Environmental Science and Pollution Research, 29(9), pp.12890-12910.
Amore, M.D. and Bennedsen, M., 2016. Corporate governance and green innovation. Journal of Environmental Economics and Management, 75, pp.54-72.
Mbanyele, W., Huang, H., Li, Y., Muchenje, L.T. and Wang, F., 2022. Corporate social responsibility and green innovation: Evidence from mandatory CSR disclosure laws. Economics Letters, 212, p.110322.
Author Response
Dear and reviewers:
The authors are very grateful to the editor and the reviewers for their constructive and encouraging comments that is helpful to guide the revision of our work. According to the valuable amendments and suggestions put forward to us! At the first time, we carefully revised and dealt with these opinions and suggestions one by one, and all opinions and suggestions have been seriously implemented. Please check it out!
For the convenience of the editor and the reviewers, we present the comments in italic and our response in the standard form. And we marked up using the “Track Changes” function.
Thank you very much for your recognition of the paper and the author's efforts. Your valuable suggestions are very important and we need to work harder in the future. In the future, we will show more and more excellent academic research achievements of our team to experts and scholars.
Comment #1:
Writing issues: For example the opening sentence "Under the background of the report of the 19th CPC National Congress clearly 40 pointed out that we should unswervingly implement the five new development concepts" . Who is being referred by "we"? Is this a quote or ?
Response: Thank you for your valuable suggestions. "We" here means "the party and state people", this is a quote , which we have updated in the manuscript (Page 2, lines 44).
Comment #2:
The authors need to improve their in-text citation style. For example, in the second paragraph, the authors just put a block of references (3-19) . The authors need to insert the cites next to the related explanation, this makes it easier for the readers.
Response: Thank you for your constructive suggestions. Following your suggestions, we have changed our in-text citation style,and insertd the cites next to the related explanation.
Comment #3:
There is a need to add other references on environmental regulation and green innovation. For example, the authors write "Foreign empirical research mainly focuses on the green innovation behavior at the industry and enterprise level..." but there is no reference to a foreign research article on green innovation. They can add Mbanyele et al., (2022) as another reference. They can supplement the three references below with others.
Response: Thank you for your valuable suggestions. Following your suggestions, we have added some foreign literature on environmental regulation and green innovation. And extensive additions and revisions to the references. Moreover, we have adopted your recommended references, thank you!
Author Response
Dear reviewers:
The authors are very grateful to the editor and the reviewers for their constructive and encouraging comments that is helpful to guide the revision of our work. According to the valuable amendments and suggestions put forward to us! At the first time, we carefully revised and dealt with these opinions and suggestions one by one, and all opinions and suggestions have been seriously implemented. Please check it out!
For the convenience of the editor and the reviewers, we present the comments in italic and our response in the standard form. And we marked up using the “Track Changes” function.
Comment #1:
Title、Abstract、Keywords
Response: Thank you for your valuable suggestions. Following your suggestions, we have made the appropriate changes and additions.
Comment #2:
Introduction:
Response: Thank you for your constructive suggestions. We are very sorry for the inconvenience caused to your reading. Following your suggestions, we have done a lot of work and effort on the introductory section. The background, importance, motivation for writing, summary of the positive and negative aspects of the literature, points of contradiction, research gaps, purpose, contributions, and structure of the article were adjusted and revised, and have been updated in the manuscript.
Comment #3:
Literature Review
Response: Thank you for your valuable suggestions. We apologize for the omission of the literature review section. We have made a supplement, please refer to (Lines 176-265, page 5-7).
Comment #4:
Discussion: Please have a discussion section before the conclusion
Response: Thank you for your constructive suggestions. Following your suggestions, we supplement the discussion part of the article and discuss it on the basis of theoretical value and policy enlightenment. Has been updated in the manuscript.(Lines 847-965, page 24-26).
Comment #5:
Future research
Response: Thank you for your valuable suggestions. Following your suggestions, we have added future studies and the limitations of this paper, which have been updated to the manuscript (Lines 1021-1037, page 28).
Reviewer 3 Report
The paper entitled “Analysis on Spatial -temporal characteristics and Influencing Factors of Green Innovation Efficiency in Three Major Urban Agglomerations in China” aim is to measure the green innovation efficiency of each urban agglomeration, and a spatial model is established to explore the spatio-temporal evolution trend and spatial effect characteristics of green innovation efficiency. To do so, the SBM-DEA efficiency model is used.
The paper is interesting and well written.
However, the following amendments are required before the paper can be published.
1. The authors must clearly explain why they have selected the specific areas of China referred in the methodology, instead of others. Furthermore it must be explained whether these areas can provide results that can be used for more cases as well.
2. Furthermore, the authors must clearly point out the paper’s novelty and contribution in the introduction section.
3. Moreover, the development of some research aims or, ideally, of some research hypotheses would facilitate the understanding of the paper’s aims, results and contribution.
4. The paper lacks of a literature review section. Instead of adding one, the authors should enrich the theoretical part of the introduction section. To do so, the following papers could be referred:
a. Skordoulis, M., Ntanos, S., Kyriakopoulos, G. L., Arabatzis, G., Galatsidas, S., & Chalikias, M. (2020). Environmental innovation, open innovation dynamics and competitive advantage of medium and large-sized firms. Journal of Open Innovation: Technology, Market, and Complexity, 6(4), 195.
b. Brilhante, O., & Klaas, J. (2018). Green city concept and a method to measure green city performance over time applied to fifty cities globally: Influence of GDP, population size and energy efficiency. Sustainability, 10(6), 2031.
c. Salehibarmi, M., Rezaei, A. A., & Noori Kermani, A. (2018). The environmental performance evaluation of tehran municipality based on the green city indicators. Urban Management Studies, 10(33), 1-15.
d. Skordoulis, M., Kyriakopoulos, G., Ntanos, S., Galatsidas, S., Arabatzis, G., Chalikias, M., & Kalantonis, P. (2022). The Mediating Role of Firm Strategy in the Relationship between Green Entrepreneurship, Green Innovation, and Competitive Advantage: The Case of Medium and Large-Sized Firms in Greece. Sustainability, 14(6), 3286.
e. Paschek, F. (2015). Urban sustainability in theory and practice-circles of sustainability. The Town Planning Review, 86(6), 745.
f. Riffat, S., Powell, R., & Aydin, D. (2016). Future cities and environmental sustainability. Future cities and Environment, 2(1), 1-23.
g. Tanguay, G. A., Rajaonson, J., Lefebvre, J. F., & Lanoie, P. (2010). Measuring the sustainability of cities: An analysis of the use of local indicators. Ecological indicators, 10(2), 407-418.
5. A paragraph of future research directions must be added.
6. A paragraph of limitations must be added as well. Some limitations are referred in the analysis. These limitations must be explained. The remedies used for these limitations must analyzed as well.
Author Response
Dear reviewers:
The authors are very grateful to the editor and the reviewers for their constructive and encouraging comments that is helpful to guide the revision of our work. According to the valuable amendments and suggestions put forward to us! At the first time, we carefully revised and dealt with these opinions and suggestions one by one, and all opinions and suggestions have been seriously implemented. Please check it out!
For the convenience of the editor and the reviewers, we present the comments in italic and our response in the standard form. And we marked up using the “Track Changes” function.
Thank you very much for your recognition of the paper and the author's efforts. Your valuable suggestions are very important and we need to work harder in the future. In the future, we will show more and more excellent academic research achievements of our team to experts and scholars.
Comment #1:
The authors must clearly explain why they have selected the specific areas of China referred in the methodology, instead of others. Furthermore it must be explained whether these areas can provide results that can be used for more cases as well.
Response: Thank you for your valuable suggestions. Following your suggestions, we have explained in detail why these three urban agglomerations were chosen in terms of importance, functional positioning, representativeness, etc. As well as studies for these three major urban agglomerations are of great importance both nationally and internationally. It has been updated in the manuscript. (Lines 290-325, page 7-8; Lines 1012-1020, page 27 ).
Comment #2:
Furthermore, the authors must clearly point out the paper’s novelty and contribution in the introduction section.
Response: Thank you for your constructive suggestions. Following your suggestions, we have added the paper’s novelty and contribution in the introduction section. (Lines 136-164, page 4).
Comment #3:
Moreover, the development of some research aims or, ideally, of some research hypotheses would facilitate the understanding of the paper’s aims, results and contribution.
Response: Thank you for your valuable suggestions. Following your suggestions, we have added research aims to show our work and findings more clearly. (Lines 266-287, page 7).
Comment #4:
The paper lacks of a literature review section.
Response: Thank you for your constructive suggestions. We apologize for the omission of the literature review section. Following your suggestions, we have made a supplement, and to enrich the theoretical part of the introduction section, please refer to (Lines 176-265, page 5-7). And extensive additions and revisions to the references. Moreover, we have adopted your recommended references, thank you!
Comment #5:
A paragraph of future research directions and limitations must be added
Response: Thank you for your valuable suggestions. Following your suggestions, we have added future research directions and the limitations of this paper, which have been updated to the manuscript (Lines 1021-1037, page 28).
Reviewer 4 Report
There is no need to show equations for DEA analysis when we have software for that.
The method Is highly subjective and expert judgement, so it would be more valuable to explain input variables in the model instead of these absolutely unnecessary formulae.
Only a fraction of patents do become an innovation as you defined it. Why not use some database tracking manufacturing innovations? I’m sure in China there is plenty of statistics.
Page 4-5 – you describe variables you use but not the source of that data so it is a bit unusual not to disclose where from you got them e.g. profits on new products?
The choice of indexes in Table 1. Input-output indicators and descriptive statistics of green innovation efficiency, have nothing to do with variables you are describing that you will be using
I don’t see the purpose of 3.3. Exploratory spatial data analysis (ESDA) – absolutely numbers have no sense when we don’t know what entered the model, and you already displayed graphic analyses and that it more than enough – but we also don’t know which variables you used form Statistical yearbooks you are referring to.
I absolutely do not understand why your already complex subject you include services on page 13. With no referencing.
Second, it is generally believed that industry is a
400 high energy-consuming and high-emission industry, while the service industry is a
401 smokeless economy and green industry, and regional green innovation mainly depends
402 on the service industry.
Page 13, lines 404-418 - pure speculation not grounded in any research or at least not cited.
Page 14 lines 419-442
We already know that government initiatives have consequences, that’s nothing new. You again forgot to cite previous studies. Now, all of the sudden you talk about education and importance of education on innovation and you show variables in table 13 that you use to prove – not sure what- which have again nothing to do with education and consequently innovation.
Page 16 line 482
The level of economic development. (1) The regression coefficient of per capita in- 483 come level is positive and passed the significance test of 1%, indicating that it has a posi- 484 tive impact on the efficiency of green innovation. – where do we actually see that? The rest is also unclear. Name the column and line where we see that.
Page 16 line 496
Generally speaking, the service in- 496 dustry represented by the tertiary industry has more characteristics of green innovation 497 development than the manufacturing industry.
Would you please explain what is green innovation in services since you decided to include them too, and where exactly we see that?
Page 18 Conclusion and enlightenment - usually we say discussion
And it is usually comparison to previous literature
Author Response
Dear reviewers:
The authors are very grateful to the editor and the reviewers for their constructive and encouraging comments that is helpful to guide the revision of our work. According to the valuable amendments and suggestions put forward to us! At the first time, we carefully revised and dealt with these opinions and suggestions one by one, and all opinions and suggestions have been seriously implemented. Please check it out!
For the convenience of the editor and the reviewers, we present the comments in italic and our response in the standard form. And we marked up using the “Track Changes” function.
Thank you very much for your recognition of the paper and the author's efforts. Your valuable suggestions are very important and we need to work harder in the future. In the future, we will show more and more excellent academic research achievements of our team to experts and scholars.
Comment #1:
The method Is highly subjective and expert judgement, so it would be more valuable to explain input variables in the model instead of these absolutely unnecessary formulae.
Response: Thank you for your valuable suggestions. Following your suggestions, we have added detailed explanations and descriptions of the input and output variables. Related updates have been submitted to the manuscript. (Lines 397-484, page 10-12).
Comment #2:
Only a fraction of patents do become an innovation as you defined it. Why not use some database tracking manufacturing innovations? I’m sure in China there is plenty of statistics.
Response: Thank you for your valuable suggestions. As the manufacturing industry involves a wide range of areas, our research can not be very comprehensive, the lack of full excavation of this area of research, we will increase the in-depth study of manufacturing innovation in the future. However, our research tracks the statistical data of "large and medium-sized industrial enterprises" and "industrial enterprises above scale".
Comment #3:
Page 4-5 – you describe variables you use but not the source of that data so it is a bit unusual not to disclose where from you got them e.g. profits on new products?
Response: Thank you for your valuable suggestions. We are very sorry for that in this manuscript and the inconvenience caused to your reading. Following your suggestions, we have supplemented and explained the source of the data and updated it in the manuscript. (Lines 473-483, page 12). e.g. The profits on new products date comes from (Lines 438-444, page 11).
Comment #4:
I don’t see the purpose of 3.3. Exploratory spatial data analysis (ESDA) – absolutely numbers have no sense when we don’t know what entered the model, and you already displayed graphic analyses and that it more than enough – but we also don’t know which variables you used form Statistical yearbooks you are referring to.
Response: Thank you for your valuable suggestions. We are very sorry for that in this manuscript and the inconvenience caused to your reading. First of all, we supplement and explain the variables in section 2.2.3. Secondly, we explain the data used in Section 3.3. The purpose of our section 3.3 study is to explore the global and local spatial distribution characteristics and agglomeration law of the three major urban agglomerations in China, which can better pave the way for the study of the spatial effect of section 3.4, and updated in the manuscript.(Lines 356-386, page 9-10; Lines 589-624, page 16-17 ).
Comment #5:
I absolutely do not understand why your already complex subject you include services on page 13.
Response: Thank you for your valuable suggestions. Here we mainly want to express the tertiary sector, due to a mistake to say services and we have carefully checked and corrected errors. We are very sorry for the mistakes in this manuscript and the inconvenience caused to your reading. (Lines 674-683, page 19).
Comment #6:
We already know that government initiatives have consequences, that’s nothing new. You again forgot to cite previous studies. Now, all of the sudden you talk about education and importance of education on innovation and you show variables in table 13 that you use to prove – not sure what- which have again nothing to do with education and consequently innovation.
Response: Thank you for your valuable suggestions. Following your suggestions, we have made additions to the literature with citations. (Lines 699-717, page 20).
Comment #7:
referencing
Response: Thank you for your valuable suggestions. Following your suggestions, we have made additions to the literature with citations. And extensive additions and revisions to the references.
Comment #8:
The level of economic development. (1) The regression coefficient of per capita in- 483 come level is positive and passed the significance test of 1%, indicating that it has a posi- 484 tive impact on the efficiency of green innovation. – where do we actually see that? The rest is also unclear. Name the column and line where we see that.
Response: Thank you for your valuable suggestions.We are very sorry for that in this manuscript and the inconvenience caused to your reading. Our results are in Table 5.
Comment #9:
Conclusion and enlightenment - usually we say discussion. And it is usually comparison to previous literature
Response: Thank you for your valuable suggestions. Following your suggestions, we supplement the discussion part of the article and discuss it on the basis of theoretical value and policy enlightenment. Has been updated in the manuscript.(Lines 847-965, page 24-26).
Round 2
Reviewer 3 Report
The authros have rigorously reviewed the paper based on the reviewers' comments and suggestion. Thus, the paper can be accepted for publication in present form.
Reviewer 4 Report
This time, authors made an effort to clarify more their work. Well done.